

# Assimilation of MODIS Dark Target and Deep Blue observations in the dust aerosol component of NMMB/BSC-CTM version 1.0

Enza Di Tomaso[1], Nick A. J. Schutgens[2], Oriol Jorba[1], and Carlos Pérez García-Pando[3,4]

[1]Earth Sciences Department, Barcelona Supercomputing Center, Spain
[2]Atmospheric, Oceanic and Planetary Physics, University of Oxford, United Kingdom
[3]NASA Goddard Institute for Space Studies, New York, USA
[4]Department of Applied Physics and Applied Math, Columbia University, New York, USA

*Correspondence to:* Enza Di Tomaso (enza.ditomaso@bsc.es)

**Abstract.** A data assimilation system has been developed for the chemical transport forecast model NMMB/BSC-CTM, with a focus on mineral dust, a prominent type of aerosol. Before this work, the system did not have an aerosol data assimilation capability and dust was produced uniquely from model estimated surface emission fluxes. As emissions are recognized as a major factor limiting the

accuracy of dust modelling, remote sensing observations from satellites have been used to improve the description of the atmospheric dust load in the model. An ensemble-based Kalman filter technique (namely the Local Ensemble Transform Kalman Filter - LETKF) has been utilized to optimally combine model background and satellite retrievals. Our implementation of the ensemble is based on known uncertainties in the physical parametrizations of the dust emission scheme. We have consid-

ered for assimilation satellite Aerosol Optical Depth (AOD) at 550 nm retrieved from measurements of top-of-atmosphere reflectances by the Moderate Resolution Imaging Spectroradiometer (MODIS) sensor on-board the NASA Aqua and Terra satellites, after applying a mineral dust filter. In particular we have assimilated two MODIS Level 3 AOD products: the U.S. Naval Research Laboratory (NRL) and University of North Dakota AOD, which is available over land and ocean, with the exclusion of

bright reflective surfaces and is based on the MODIS Dark Target Collection 5 Level 2 product, and the MODIS Deep Blue Collection 6 AOD, which is available over land including bright arid surfaces, such as deserts. Data assimilation experiments using the LETKF scheme have been evaluated against observations from the Aerosol Robotic Network (AERONET) of ground-based stations and against MODIS satellite retrievals. Experiments showed that AOD retrievals using the Dark Target

algorithm can help NMMB/BSC-CTM to better characterize atmospheric dust. This is particularly true for the analysis of the dust outflow in the Sahel region and over the African Atlantic coast. The additional assimilation of retrievals based on the Deep Blue algorithm has a further positive impact in the analysis downwind from the strongest dust sources of Sahara and in the Arabian peninsula. An analysis-initialized forecast performs better (lower forecast error and higher correlation with ob-



servations) than a standard forecast, with the exception of underestimating dust in the long-range Atlantic transport and degradation of the temporal evolution of dust in some regions after day 1. Particularly relevant is the improved forecast over Sahara throughout the forecast range thanks to the assimilation of Deep Blue retrievals over areas not easily covered by other observational datasets.

The present study on mineral dust is a first step towards data assimilation with a complete aerosol chemical transport model that includes multiple aerosol species.

## 1 Introduction

Among the different aerosol species, mineral dust is one of the main components of the atmospheric aerosol loading and is of great interest for a variety of reasons. Mineral dust plays an important role in the earth's energy balance and has a relevant impact on economical activities, on the ecosystem, on health, as well as on weather and climate (Knippertz and Stuut, 2014). The strong dust storms occurring near emission sources constitute a big hazard to air traffic and road transport as they can reduce the visibility down to few meters. However dust does not affect only local economies: because of its transport over thousands of kilometres, it has an impact from local to global scales. Dust deposition provides nutrients to continental and marine ecosystems. African dust for example has a role in fertilization of the Amazon rainforest (Yu et al., 2015), while dust deposition over oceans has implication on sea biogeochemistry as the iron contained in the dust particles is a nutrient for phytoplankton, whose photosynthetic activity in turn affects the carbon cycle (Jickels et al., 2005). Dust has health implications both close and far from sources. For example, studies have shown the usefulness of dust aerosol climatologies to predict part of the year-to-year variability of the seasonal incidence of meningitis in Niger (Pérez García-Pando et al., 2014), while particulate matter measurements taken in areas far from sources show that Saharan dust outbreaks have adverse effects of cardiovascular and respiratory conditions (Mallone et al., 2011; Morman and Plumlee, 2013; Pandolfi at al., 2014). Mineral particles perturb the earth-atmosphere's radiation budget through their interaction with the short-wave radiation, through scattering and absorption, and long-wave radiation, through absorption and re-emission. Due to this redistribution of the energy, dust aerosols can have a strong impact on atmospheric processes at short (weather) and long (climate) term periods while they can affect atmospheric circulations at large spatial scales (e.g. Asian monsoons; Lau et al. (2006)). Furthermore, this can generate feedback processes since changes in weather and climate can in turn lead to changes in the dust cycle.

Different types of ground-based (e.g., Kim et al., 2011; Pey at al., 2013) and space-borne (e.g., Kaufman et al., 2005; Luo et al., 2015) observations have been utilized to describe the variability of atmospheric dust. However, due to either insufficient spatial representativeness or accuracy, the spatio-temporal features of dust aerosols are not fully captured by the current observing system. Neither do models accurately describe atmospheric and surface dust concentrations (Huneeus et al.,



2011). High uncertainties are also in our knowledge of the optical and micro-physical properties of
dust, and in our representation of its vertical structure. The latter has implication on the radiation's
budget and transport. On the other hand, an accurate quantification of dust's spatial and temporal
distribution is key in correctly characterizing the effect that it has on the earth's energy balance,
as well as in improving the skill of forecasting its concentrations in the atmosphere as well as in

forecasting the weather (Pérez García-Pando at al., 2006; Grini et al., 2006; Chaboureau et al., 2011).

Regional and global centres, predicting the most important aerosol species or dust only, participate
to different model inter-comparison initiatives like the Aerosol Comparisons between Observations
and Models (AeroCom; Tsigaridis et al., 2007) project, the International Cooperative for Aerosol
Prediction (ICAP; Sessions et al., 2015) initiative and the WMO Sand and Dust Storm Warning Ad-

visory and Assessment System (SDS-WAS; Terradellas et al., 2015). Multi-model ensemble spreads
give an indication of large uncertainties in the modelling schemes and confirm the need of a bet-
ter characterization of aerosols. Relatively recently the atmospheric composition community has
resorted to data assimilation (DA) for a better characterization and prediction of atmospheric con-
stituents such as aerosols and trace gases (Bocquet et al., 2015). Though their dynamic is mainly

driven by forcings such as emissions, recent studies showed that the usage of observations through
data assimilation has improved significantly the accuracy of short-term forecast and the global mon-
itoring of both aerosols and trace gases (Benedetti et al., 2009; Elbern and Schmidt, 2001). Since the
first experiments in the early 2000s, the assimilation of aerosol observations is now operational in
some of the main aerosol forecasting centres. Zhang et al. (2014) have highlighted in particular the

importance of a combined assimilation of satellite products for aerosol forecast.

The Earth Sciences Department of the Barcelona Supercomputing Centre (ES-BSC) is implement-
ing a gas-aerosol module able to predict atmospheric composition at different spatial and temporal
scales within the state-of-art meteorological model NMMB (Non-hydrostatic Multi-scale Model on
the B grid; Janjic and Gall, 2012). This modelling system is known as the NMMB/BSC-Chemical

Transport Model (NMMB/BSC-CTM). We report here on the extension of NMMB/BSC-CTM with
a data assimilation functionality using satellite aerosol optical depth. NMMB/BSC-CTM version
1.0, as in Pérez García-Pando et al. (2011), considers dust only but other aerosols are being imple-
mented (Spada et al., in prep). The focus of this work on mineral dust is justified by the operational
services provided by the NMMB/BSC-CTM. This model provides an operational dust forecast for

the Barcelona Dust Forecast Centre under an initiative of the World Meteorological Organization.
It participates in the multi-model dust ensemble of the aforementioned ICAP initiative, providing
daily global dust forecast up to 120 hours. It also provides daily regional forecast up to 60 hours to
the WMO SDS-WAS system. Before this work, the system did not have an aerosol data assimila-
tion capability and dust was produced uniquely from model estimated surface emission fluxes. The

present study on mineral dust is a first step towards data assimilation with a complete aerosol chem-





ical transport model that includes multiple aerosol species (not only dust but also seasalt, sulphate and organics).

Previous studies of assimilation of dust aerosol only have been conducted for the Chinese Unified Atmospheric Chemistry Environment - Dust (CUACE/Dust) forecast system (Niu et al., 2008; Wang and Niu, 2013). These studies have used variational data assimilation techniques (3D-Var) which require, in their most practical implementation, pre-calculated and constant in time model error structures. Alternatively, ensemble-based techniques use flow-dependent model error amplitudes and structures which evolve during forecast and, in theory, should be able to capture better instabilities in the background flow (Evensen, 1994; Kalnay et al., 2007).

In this work we present the coupling of NMMB/BSC-CTM with an ensemble-based technique known as Local Ensemble Transform Kalman Filter (LETKF; Hunt et al., 2007; Miyoshi and Ya-mane, 2007). The LETKF scheme has shown to be particularly suitable for the assimilation of aerosol information since it has been observed by Anderson et al. (2003) , Shinozuka and Redemann (2011), and Schutgens et al. (2013) that aerosol fields have limited spatial correlations. The utility of ensemble-based techniques for global aerosol simulations has been shown in previous studies (Schutgens et al., 2010a; Sekiyamal et al., 2010; Rubin et al., 2016; and more recently Yumimoto et al., 2016). The main novelty in our study is the creation of the ensemble, our implementation is based on known uncertainties in the physical parametrizations of the sophisticated dust emission scheme used by the NMMB/BSC-CTM model, as well as in the use of observations particular relevant for dust applications, like MODIS Deep Blue.

The NMMB/BSC-CTM chemical transport model is described in more detail in Section 2, with emphasis on its dust module. A description of the data assimilation scheme and of the assimilated observations follows respectively in Section 3 and Section 4. We report then in Section 5 about the characteristics of the simulations that we have run, in Section 6 about the evaluation methodology that we have followed, and in Section 7 about the evaluation results of our simulation experiments. The final section concludes the paper with a summary of this development, the main results achieved, and future perspectives.

## 2 The NMMB/BSC Chemical Transport Model and its mineral dust component

The ES-BSC is implementing a new gas-aerosol module within the NMMB meteorological model from the Unites States National Centers for Environmental Prediction (NCEP). The new modelling system is known as the NMMB/BSC-CTM (Pérez García-Pando et al., 2011; Jorba et al., 2012; Spada et al., 2013), and is developed in collaboration with NCEP and other research institutions. The chemistry (aerosols included) and meteorology are fully on-line integrated. NMMB/BSC-CTM is able to work with a wide range of spatial scales thanks to its unified non-hydrostatic dynamical core, keeping consistent parametrizations at different spatial and temporal scales. Furthermore, the





dynamical core and the coupled modules are computationally highly efficient satisfying current and projected operational requirements. The rest of this section will briefly describe some characteristics of the dust component of the NMMB/BSC-CTM, with particular focus on the emission scheme.

The dust emission scheme implemented in the NMMB/BSC-CTM follows the empirical relation-
135 ship of Marticorena and Bergametti (1995) and Marticorena et al. (1995) according to which the vertical dust flux is proportional to the horizontal sand flux. The horizontal to vertical flux ratio reflects the availability of dust in four soil populations (clay, silt, fine/medium sand, and coarse sand) (Tegen et al., 2002). The horizontal sand flux is modelled as the flux of the saltating particles $H$ simulated according to White (1979) and proportional to the third power of the wind friction velocity. A
140 soil moisture-dependent threshold on the friction velocity determines the velocity above which the soil particles begin to move in horizontal saltation flux. Soil moisture effects are included following Fecan et al. (1999) through the soil moisture correction parameter included in the expression for the threshold on the friction velocity. A sectional approach is used for the transport of dust particles, i.e. the dust size distribution is decomposed in small size bins. More exactly, dust is modelled
using eight dust size bins varying from 0.1 to 10 microns, and, within each transport bin, dust is assumed to have a time-invariant lognormal distribution (Zender et al., 2003). The total vertical flux mass is distributed among the dust transport bins according to a specific dust distribution at sources. NMMB/BSC-CTM uses a distribution over sources derived from D'Almeida (1987) which assumes that the vertical dust flux is size distributed according to three lognormal background source modes.
More explicitly, the dust vertical mass flux $F_k$ in a given transport bin $k$ at each grid cell is given by

$$F_k = C\,S\,(1-V)\,\alpha\,H\sum_{i=0}^{3} m_i\,M_{i,k} \qquad\qquad k = 1,\ldots,8 \qquad\qquad (1)$$

where $S$ is a source erodibility factor defined on bare ground surfaces, representing the probability to have accumulated sediments in the given grid cell that are potential dust sources; $(1-V)$ is the grid's fraction of bare soil; $\alpha$ is the horizontal to vertical flux ratio calculated for four soil populations
classes (clay, silt, fine/medium sand, and coarse sand); $H$ is the horizontal sand flux; $M_{i,k}$ is the mass fraction of background source mode $i$ carried in a transport bin $k$ calculated following Zender et al. (2003), and weighted by specific background source mode coefficient $m_i$; and $C$ is a global tuning factor, which is meant to compensate for the uncertainty associated with the different component of $F_k$. More details about the above formulation of dust emission can be found in Pérez García-Pando
et al. (2011).

The mineral dust module has been extensively evaluated in studies at global and regional scales (Pérez García-Pando et al., 2011; Haustein et al., 2012; Huneeus et al., 2011, 2016), showing that its evaluation scores lie in the upper range of the AEROCOM model evaluation performance scores. However, these evaluation efforts confirmed, similarly to other modelling systems, different sources
of uncertainty in the NMMB/BSC-CTM dust modelling.





## 3    The data assimilation scheme

We have coupled the NMMB/BSC-CTM with the LETKF scheme (Hunt et al., 2007; Miyoshi and Yamane, 2007; Schutgens et al., 2010a) with four-dimensional extension as described in Hunt et al. (2007), in order to estimate optimal initial conditions for our dust model. The overall scheme implements an iterative approach consisting in a forecast step and state estimation step. The state estimation step combines information from mineral dust observations and a prior first-guess, or background from our model. A short-term forecast is used as background information. The background incorporates information from past observations, therefore the analysis is estimated using both current and past observations. LETKF is a development of the ensemble-based transform Kalman filter (ETKF; Bishop et al., 2001) and of the local ensemble Kalman filter (LEKF; Ott et al. , 2004), and is particularly suited to high-performance computing applications. A very attractive feature of an ensemble-based technique is the use of a flow-dependent background error covariance, which is derived from the ensemble of model states at the assimilation time, and evolves during forecast. At any given time in fact the state estimate is represented by an ensemble of system states and its uncertainty is derived from the ensemble. LETKF has the advantageous feature that it applies localization, i.e. it performs the analysis locally (at each grid point only observations within a certain distance are assimilated), allowing the global analysis to explore a much higher-dimensional space than the subspace spanned by the ensemble (whose dimensionality is limited by the number of ensemble members). Localization also reduces the effect of spurious long-range covariances, effectively eliminating them after a given distance. This is particularly suitable for the assimilation of aerosol information since, as mentioned in the introduction, it has been observed that aerosol fields have limited spatial correlations ($\sim$100 km). Schutgens et al. (2010a, b) have already shown the positive impact of assimilating aerosol ground station observations using a LETKF assimilation system for the SPRINTARS model, while Sekiyamal et al. (2010) used it to assimilated CALIOP vertical profiles in the MASINGAR model and Dai et al. (2013) used it to ingest MODIS observations in the NICAM-SPRINTARS model.

Here we discuss the basic concepts behind the LETKF algorithm, a more detailed description of the scheme can be found in Hunt et al. (2007). Consider a state vector $\boldsymbol{x}$ of the dynamic variables of a system (for our application this is dust mass mixing ratios). The mean analysis increment at a grid point is estimated as a linear combination of the background ensemble perturbations $\mathbf{X}^b$ :

$$\bar{\boldsymbol{x}}^a = \bar{\boldsymbol{x}}^b + \mathbf{X}^b \mathbf{w} \tag{2}$$

where we use the superscripts $a$ and $b$ to denote respectively the analysis and background state vector, and where the $i$th column of the matrix $\mathbf{X}^b$ is $\boldsymbol{x}^{b(i)} - \bar{\boldsymbol{x}}^b$, $\{i = 1, 2, \ldots, k\}$ with $k$ ensemble members, i.e. the difference between the $i$th ensemble forecast $\boldsymbol{x}^{b(i)}$ and the ensemble forecast mean $\bar{\boldsymbol{x}}^b$. $\mathbf{w}$ is termed the "weight" matrix specifying what linear combination of the background ensemble perturbations is added to the background mean to obtain the analysis ensemble. The "weight" matrix





is given by

$$\mathbf{w} = [\mathbf{Y}^b \mathbf{R}^{-1} \mathbf{Y}^b + (k-1)\mathbf{I}]^{-1} \mathbf{Y}^b \mathbf{R}^{-1} (\boldsymbol{y}^o - \bar{\boldsymbol{y}}^b) \qquad (3)$$

where $\mathbf{Y}^b$ is the background ensemble perturbation matrix in observation space (or background ob-
servation ensemble perturbation matrix), $\mathbf{R}$ is the observation error covariance matrix which we
assume diagonal, $\mathbf{I}$ is the identity matrix, $\boldsymbol{y}^o$ is the vector of observations and $\bar{\boldsymbol{y}}^b$ is the mean back-
ground observation ensemble. The background observation ensemble is obtained applying the ob-
servation operator $h(\cdot)$ to the ensemble forecast members $\boldsymbol{x}^{b(i)}$, i.e. $\boldsymbol{y}^{b(i)} = h(\boldsymbol{x}^{b(i)})$.

LETKF uses R-localization, i.e. the localization is performed in the observation error covariance
matrix, making the influence of an observation on the analysis decay gradually toward zero as the
distance from the analysis location increases. To achieve this, the observation error is divided by a
distance dependent function that decays to zero with increasing distance: $e^{-\frac{dist^2}{h^2}}$, where $dist$ is the
distance between an observation and the model grid in which the analysis is calculated, and $h$ is
horizontal localisation factor.

## 3.1 Ensemble perturbations

We run the data assimilation scheme under an imperfect model scenario assumption: each ensemble
member is run with a different perturbation of uncertain model parameters in the dust emission
scheme. In dust modelling, the emission source term is a particularly large contributor to model error
(Huneeus et al., 2011). In the case of NMMB/BSC-CTM one of the component to the uncertainty in
the emission term has been identified for example in the vertical flux distribution at sources (Gama
et al., 2016).

The model ensemble is created perturbing the vertical flux of dust in each of the eight dust bins. As
described in Section 2, NMMB/BSC-CTM follows a sectional approach for dust, i.e. the size distri-
bution is decomposed in small size bins. This is equivalent to perturbing the total vertical flux as well
as its size distribution at sources. The perturbations are extracted imposing some physical constraint:
correlated noise is used across the bins so that noise correlation decreases with increased difference
of the normalized cubic radius among the bins; the noise is applied multiplicatively and has mean 1
and standard deviation of 30% of the unperturbed value in each bin; and the final distribution is uni-
modal. Figure 1 shows how the vertical flux is perturbed in our ensemble simulations. Additionally,
we have perturbed the threshold friction velocity for dust emission by extracting a multiplicative
random factor from a normal distribution with mean 1 and spread 0.4. This considers the uncertainty
of the model with respect to both surface winds and soil humidity. At low resolution, model surface
winds are typically underestimated over dust sources. Also, the model uses the scheme of Fecan
et al. (1999) to calculate the increase of the threshold friction velocity with soil humidity, which is
typically overestimated in arid regions (Haustein et al., 2015). The spin-up period for the ensemble
ensures that perturbations applied at the sources propagate everywhere in the globe. For this reason





at this first stage of development of our ensemble system we did not consider necessary a combined meteorology and source perturbation. The structure of our source perturbations is temporally and spatially constant.

### 3.2 Observation operator

Our state vector is the dust mass mixing ratios. Therefore an observation operator is needed to map the ensemble mean state vector into the observation space. The simulated AOD at wavelength $\lambda$ is calculated at a given observation location according to the following linear operator:

$$AOD_\lambda = \sum_{b=1}^{8} \frac{3}{4\rho_b r_b} M_b Q_{\lambda b} \tag{4}$$

where $\rho_b$ is the particle mass density, $r_b$ is the effective radius, $M_b$ is the dust column mass loading calculated from each dust bin mixing ratio, and $Q_{\lambda b}$ is the extinction efficiency factor which is calculated for using the Mie scattering theory assuming dust spherical, non soluble particles, and, within a bin, a lognormal distribution for dust with geometric radius of $0.2986$ $\mu m$ and standard deviation of $2.0$.

When using in the state vector the total mass mixing ratio, as we will explain in Section 5, an ensemble averaged extinction efficiency is calculated as in Schutgens et al. (2010b) as an average of the extinction efficiency of the individual bins weighted by the bin mixing ratios.

Hereafter, when we will use the term AOD without specifying the wavelength, we imply that we refer to aerosol optical depth at a wavelength of 550 nm, which is the most commonly reported value in the literature.

## 4 Observational data

### 4.1 MODIS Dark Target and OMI

We consider for assimilation the MODIS Level 3 AOD product produced by the U.S. NRL and University of North Dakota ((hereafter called NRL MODIS). The NRL MODIS product is produced for the purpose of assimilation into aerosol transport models (Zhang and Reid, 2006; Hyer et al., 2010; Shi et al., 2011), post-processing the Level 2 MODIS Dark Target product from the so-called Collection 5 (Remer et al., 2008; Levy et al., 2007a, b), and is available both over land and ocean. The MODIS Level 2 product is an average of the 1 km by 1 km retrievals (at nominal resolution) generated by the Dark Target algorithm applied to top-of-atmosphere reflectances observed by the MODIS sensor on-board of the NASA polar-orbiting satellites Terra and Aqua. The NRL MODIS Level 3 product is filtered and corrected in order to eliminate outliers and gross systematic biases, spatially aggregated into a 1 by 1 degree mesh in order to avoid the assimilation of sub-grid features, and an error is estimated for each observation. The product is generated every six hours at 0, 6, 12,



18 UTC and is based on MODIS Level 2 observations in a 6 hour interval around those times. The
retrieval errors estimated by NRL/University of North Dakota were used for the observation errors.
They include the instrumental error variance and the spatial representation error variance. Following
the approach in Zhang et al. (2008), we assume uncorrelated observation errors. These observations
pertain to the total atmospheric aerosol column, not just to dust AOD. The selection of observa-
tions in dust-dominated conditions is performed using retrievals of Ångström Exponent (AE) from
the original MODIS Level 3 product (Hubanks et al., 2008), for information about the size of the
particles, and using retrievals of Aerosol Absorbing Index (AAI) from the Ozone Monitoring Instru-
ment (OMI) sensor (Torres et al., 2007), for information about the absorption characteristics of the
particles. Ångström Exponent (AE) values are based on quality assurance-weighted 470 and 660nm
optical depths over land, and 550 and 865nm optical depths over sea. Observations are selected when
daily MODIS Aqua or Terra products contain a value for AE<0.75 and daily OMI products contain
value for AAI>1.5. Figure 2 shows an example for the NRL MODIS Level 3 product for a day of
May 2007 after the filter for dust-dominated conditions is applied.

## 4.2 MODIS Deep Blue

The MODIS Dark Target product does not provide information over very bright reflective surfaces,
including deserts, as the retrieval algorithm assumes low surface albedo. We consider the assimila-
tion of MODIS Deep Blue Level 3 daily AOD product from Collection 6 whose algorithm retrieves
AOD also over bright arid land surfaces, such as deserts. The Collection 6 product covers all cloud-
free and snow-free surfaces, and can be potentially very useful for mineral dust applications as it is
able to provide observational constraint close to dust sources. The Deep Blue algorithm uses top-of-
atmosphere reflectances at $412$ and $470\,nm$. In the presence of heavy dust load also the reflectance at
$650\,nm$ is used. The algorithm exploits the fact that, over most surfaces, darker surface and stronger
aerosol signal is seen in the blue wavelength range than at longer wavelengths. The quality of the
MODIS Deep Blue AOD product is improved in Collection 6 compared to Collection 5, as work of
Sayer et al. (2014), based on Level 2 retrievals, showed. Similar findings, for the northern African
and Middle East deserts, were reported by Gkikas et al. (2015b), who used Level 3 retrievals over
the period 2002-2014.

We have applied to this product the same filter for dust-dominated conditions described in Section
4.1. In addition we have masked out Level 3 retrievals obtained with less than 30 Level 2 retrievals,
since Gkikas et al. (2015a) showed that the agreement between MODIS-AERONET is improved
when the sub-pixel spatial representativeness is increased. The MODIS Deep Blue observations are
not corrected for possible systematic biases, however, we are aware that for future applications we
should address any possible bias in the product. It is important to notice that the Level 3 product is an
aggregation of Level 2 retrievals that is produced using the highest quality retrievals (i.e. retrievals
with quality assurance flag value 3). Furthermore, we have applied a quality control on all the as-



similated observations based of normalized first-guess departures. As proxy for the normalization
       factor, we have used the standard deviation of first-guess departures.

       A study by Sayer et al. (2014) shows that highest quality data have an absolute uncertainty of
       approximately $(0.086 + 0.56 AOD_{550})/AMF$, where AMF is the geometric air mass factor with a
       typical AMF value of 2.8. We have used this quantification of the uncertainty for the Level 3 product.
Furthermore, we have defined the representation component of the error as the standard deviation of
       the values used in the aggregated product. Although a more accurate treatment for the representation
       error could be envisaged following for example the approach of van Leeuwen (2014), we deem small
       the impact that our approximation has on the analysis. Figure 3 shows an example for the MODIS
       Deep Blue Collection 6 Level 3 product for a day of May 2007 after the filter for dust-dominated
conditions is applied.

### 4.3   AERONET

       For validation purposes we have used observations from the ground-based stations of the global
       Aerosol Robotic Network (AERONET; Holben et al., 1998) of direct-sun photometers. These ob-
       servations have not been assimilated in our test simulations. In particular, we have used their re-
trievals of column-integrated aerosol optical depth from direct-sun photometric measurements. The
       retrievals are obtained observing the extinction of direct solar radiation due to the presence of
       aerosols in the atmosphere. For this reason AERONET retrievals are not available under cloudy
       sky conditions and during night-time. These observations suffer of a relatively sparse spatial cover-
       age but are very valuable for validation purposes as their uncertainty on these retrievals is estimated
to be between $0.01$ and $0.02$. Several studies have in fact used the AERONET data for validation
       purposes, or for the correction of biases in satellite measurements (Zhang and Reid, 2006; Hyer et
       al., 2010). We considered cloud-screened and quality-assured (Level 2.0) direct-sun AOD retrievals
       between $440$ and $870\ nm$. AERONET AOD at $550\ nm$ was obtained using the Ångström law.

### 5   Numerical simulation set up

We have run a set of different experiments (listed in Table 1): a control experiment to produce a
       5-day forecast (hereafter called Control experiment) with the same operational configuration (but
       at a coarser resolution) and version that provides daily global forecast to the aforementioned ICAP
       multi-model ensemble, and which is initialized for dust from the previous day 24 hour forecast
       (FC+24). Assimilation experiments were run with NRL MODIS AOD (hereafter called DA-NRL
experiment) and with NRL MODIS AOD and MODIS Deep Blue AOD (hereafter called DA-NRL-
       DB experiment) with a preprocessing to the observations as described in Section 4. Additionally,
       we have run also free ensemble simulations without assimilating any observation (hereafter called
       ENS-free-run). We have also run a 5-day forecast experiment initialized from the analysis produced





by the DA-NRL-DB experiment (hereafter called AN-initialized experiment) in order to evaluate
the impact of the analysis on the forecast. The Control experiment was run for five months in the
spring/summer period of 2007 (from 1 April to 31 August 2007) starting from a cold start for dust
and with a spin up period of one month (April 2007) which is excluded from the analysis of the
results. Also the ensemble is spun up before data assimilation is applied.

We use a 24-hour assimilation window and observations are considered for assimilation at four
time slots within the window, at $0$, $6$, $12$ and $18$ UTC. The system uses as first-guess a 1-day forecast
with output every 6 hours. Simulated observation and background departures are calculated at each
time slot. The time slots are exactly the ones corresponding to the times in which NRL MODIS
AOD observations are available. We are using the LETKF implementation with a four-dimensional
extension as described in Hunt et al. (2007). The state vector comprises of the mixing ratio at all
350 the time slots considered and so does the observation AOD vector. Background observation means
$\bar{\mathbf{y}}_j$ and perturbation matrices $\mathbf{Y}_j$ are formed at the various time slots $j$ when the observations are
available. They are then vertically concatenated to form a combined background observation mean
$\bar{\mathbf{y}}$ and perturbation matrix $\mathbf{Y}$. $\bar{\mathbf{y}}$ and $\mathbf{Y}$ are used for the calculation of a weight matrix $\mathbf{w}$ using the
standard LETKF, i.e. we calculate a single $\mathbf{w}$ based on all innovations throughout the day. This same
$\mathbf{w}$ is then applied to the state vector at different times throughout the assimilation window.

We have tuned different aspects of the data assimilation system including testing the number of
ensemble members, different perturbations of the ensemble, and a different state vector for the con-
trol variables. Using $24$ ensemble members did not produce a significant impact on the dust analysis
compared to the use of $12$ ensemble members. This could be explained with our setting of a localiza-
360 tion factor which makes the influence of an observation on the analysis decay gradually toward zero
as the distance from the analysis location increases. We have set the horizontal localization factor
to the value $1$ in all the data assimilation experiments. Covariance localization in fact effectively
eliminates background spatial correlations after a certain distance, and might have solved possible
sampling errors introduced by the low dimensionality of the 12 member ensemble compared to the
365 $24$ member ensemble. We also apply vertical localization following Miyoshi and Yamane (2007) ap-
proach of localizing the error covariance vertically for radiance assimilation. The observation error
is divided by the square of the model AOD normalised sensitivity function.

We have tested the usage of different perturbations of the dust emission scheme: a perturbation
of the mass vertical flux per dust bin, or a the perturbation of both the mass vertical flux and the
370 threshold on the wind friction velocity. As we show in the next section, the latter configuration was
deemed better as it spans a larger space of possible system states.

We have tested two different options for the state vector: a control variable consisting of the mix-
ing ratio of eight individual dust bins or the total dust mixing ratio defined as the sum of the eight
dust bins at each grid point and for all the vertical levels. In the latter case the mixing ratios for the
375 individual dust bin after data assimilation are determined from the background, i.e. from their rela-





tive fractions before assimilation. The observation operator is calculated using the original mixing ratio following the approach for the observation operator in Schutgens et al. (2010b). The tests that we have performed show that representing individually the bins in the state vector does not have any significant impact on the analysis, while it increases the computational cost of the assimilation compared to using the total mixing ratio. Moreover, the use of a bulk approach is common in systems assimilating total AOD values as the observations are not able to fully constrain the individual bin profiles. We should note that this choice of state vector makes still meaningful the creation of the ensemble perturbing the vertical flux for the individual bins, as this allows us to express in the background the uncertainty in the size distribution at sources, and to span a larger space of possible system states.

In the next section we show the results of assimilating NRL MODIS NRL and MODIS Deep Blue observations using 12 ensemble members obtained perturbing the mass vertical flux per bin at sources together with the threshold on the wind friction velocity, as described in Section 3.1, and using the total dust mixing ratio as analysis variable in the state vector. All simulations were run on a global domain with 40 hybrid pressure-$\sigma$ layers, 5 hPa top pressure, and a horizontal resolution of 2.8 by 2 degree. The NCEP final analysis at 1 by 1 degree at 0 UTC were used to initialize the meteorology at every forecast run.

## 6  Methodology for the evaluation of the simulations

The evaluation of the simulations is done in three stages: (a) an internal check of the data assimilation system; (b) evaluation of the analysis using as reference independent observations; (c) evaluation of a 5-day forecast with and without analysis initialization using as reference independent observations.

The consistency of the data assimilation system is checked through considerations on statistics of the ensemble, of simulation departures from assimilated observations, and of the temporal mean of assimilation increments. The ensemble mean and the coefficient of variation for the ensemble are calculated with and without data assimilation. The coefficient of variation is defined as ratio of the standard deviation of the ensemble to the ensemble mean. Additionally, statistics for first-guess (FG) and analysis (AN) departures are calculated, where departures are defined as difference between assimilated observations and simulations (first-guess or analysis), while mean increments are defined as temporal mean of differences between analysis and first-guess at the different time slots of the assimilation window.

The evaluation of analysis and forecast with respect to independent observations are performed in terms of statistics of model field errors $e_i$ from observations, where $e_i = o_i - m_i$, with index $i$ indicating an instance of observation $o_i$ and where $m_i$ is the model field in observation space, bi-linearly interpolated at the observation location. We consider the root mean square error ($RMSE$), the mean error ($BIAS$), the standard deviation of the error ($SD$), the fraction gross error ($FRGE$), and the





correlation coefficient ($CORR$) of the model AOD compared to either quality-assured (Level 2.0) AERONET or to satellite retrievals. The $FRGE = \frac{2}{n} \sum_{i=0}^{n} \left| \frac{o_i - m_i}{o_i + m_i} \right|$ is added to the most widely used set of statistics for the error as it behaves symmetrically with respect to under and over estimation without emphasizing the outliers, and is normalized to the sum of observation and simulation values. The SD of the error, though it can be derived from the other statistics, is also reported so to make more explicit the changes in the bias-free mean square error and aid the interpretation of the evaluation results. The above set of evaluation statistics are calculated for measurements from individual ground-based stations, groups of stations, regional domains observed by satellite sensors, and globally.

For AERONET AOD measurements dust-dominated conditions are identified using the approach of Basart et al. (2009) as follows: AOD is classified as 'Dust' AOD when the associated AE<0.75; we set 'Dust' AOD to 0 when the associated AE>1.3; we identify a mixed aerosol type when the associated 0.75<AE<1.3. The latter values are excluded from the validation. For satellite AOD retrievals we use the set of satellite observations quality controlled and filtered for dust-dominated conditions used in the assimilation step. We use these satellite observations to validate uniquely the forecast range following the assimilation window. We show the forecast evaluation statistics corresponding to measurements and simulations at 12 UTC only, so that they refer to an approximate equal number of pair of observations and model simulated values at each forecast lead time that we are considering.

We have identified eight regions of interest for the validation purposes in our study period, namely: Long Atlantic transport (LongAtl), Short Atlantic transport (ShortAtl), Sub-Sahel (SubSahel), Sahara (Sahara), Extended Mediterranean (ExtMediter), Middle East (MiddleEast), Central Asia (CenAsia), East Asia (EastAsia). These names do not necessary correspond to the conventional names of exact geographical locations but are meant to identify regional domains in a convenient way according to dust intrusions and to group observational stations. Most of regional domains contain ground-based stations with a minimum number of observations during the study period (stations with less than 30 'Dust' observations are discarded), with the exception of Central and East Asia. The ground-based stations are listed in Table 2, and shown in the map in Figure 4 together with regional domains used for the validation of the experiments either against ground-based or satellite observations.

## 7 Evaluation results

### 7.1 Ensemble, departure and increment statistics

We compare here the dust fields in the Control, ENS-free-run, DA-NRL and DA-NRL-DB experiment in terms of mean values and, when applicable, ensemble spread. Figure 5 shows the AOD values averaged over a month of the study period for the four above experiments. By visual inspec-





tion it can be noticed that the ensemble mean of the ENS-free-run experiment compares well with the Control experiment, which suggests that the ensemble perturbations are not altering the model mean state as defined by a standard run. The analysis clearly shows conspicuous changes in the dust field compared to the Control experiment. Figure 6 shows the coefficient of variation for AOD in the ENS-free-run and DA-NRL-DB experiment. Data assimilation clearly lowers the values of the coefficient of variation in the regions where observations are present, which indicates a reduction of the ensemble spread due to the assimilated observations. The ensemble of Figure 6 (and Figure 5) is created perturbing the emitted mass vertical flux for each dust bin and the threshold on the friction velocity generating dust horizontal flux. Creating the ensemble without perturbing the threshold on the friction velocity produces a reduced spread. See Figure 7 for this second configuration of the ensemble with ensemble mean and coefficient of variation for the ENS-free-run in the top panels, and the experiment with data assimilation in the bottom panels. Perturbing the threshold on the friction velocity has an impact on the spread also outside source regions because, as explained earlier, the spin-up period for the ensemble ensures that perturbations applied at the sources propagate everywhere. A smaller spread might translate to giving a lower weight to observations with respect to the background, therefore the ensemble with higher spread is used in the simulations described in the rest of the paper.

We evaluate in the rest of this section the assimilation experiments in terms of statistics of the departures of the analysis and first-guess from the assimilated satellite observations. Figure 8 shows for May to August 2007 first guess AOD (on the left panels) and analysis AOD (on the right panels) versus observations for the DA-NRL and DA-NRL-DB experiment. The departure statistics with respect to the two sets of observations that we have assimilated are in Table 3. In both experiments a smaller scatter and a higher correlation coefficient for the analysis indicate that the assimilation improves the agreement with observations and hence a positive sanity check of the data assimilation system. The asymmetric behaviour of all the analysis scatter plots suggests that the system is more successful in correcting too high AOD values than correcting too low AOD values, which could be due to the fact that usually we have larger observation errors and smaller ensemble spread for low AOD values. The BIAS is significant smaller than the RMSE and the RMSE improves in the analysis over the forecast. The issue of a higher BIAS in the analysis departures compared to the first-guess departures has been identified in other assimilation system (see Benedetti et al. (2009), Section 4) and might be attributed to the fact that AOD is a positive definite variable, as this provides a deviation from the Gaussianity condition in the prior which is assumed in the analysis step. A solution to this problem worth investigating in the future would consist in applying a transformation of the state variables into new variables which present Gaussian features, a procedure known as Gaussian anamorphosis (Amezcua and Van Leeuwen, 2014).

Figure 9 shows global maps of mean analysis increments, i.e. the monthly-averaged difference between analysis and short-term forecast, respectively in the case in which only NRL MODIS AOD



observations are assimilated and in the case in which also MODIS Deep Blue AOD observations are assimilated. Both experiments show non-zero systematic increments which are to be interpreted

as systematic corrections that these sets of observations are making, in particular removing mass close to sources and, to a lesser extent, adding mass in the outflow. The spatial distribution of the increments highlights the role that MODIS Deep Blue observations play in particular over the Sahara dust sources.

## 7.2 Validation of the analysis

We perform in this section a validation of the dust fields simulated either with or without data assimilation through a comparison with observations from ground-based stations that have not been assimilated for May to August 2007. We calculate the statistics for individual stations and for groups of stations. Figure 10 shows the time-series of AOD values for May to August 2007 for the Control (blue), DA-NRL (green), DA-NRL-DB (red) experiment, and for AERONET observations in

dust-dominated conditions (black) at six locations within the different regional domains of Figure 4, which are in the proximity of dust sources (Tamanrasset in Algeria), affected by short-range dust transport (Dakar in Senegal, Ilorin in Nigeria, and Hamim in the United Arab Emirates), or affected by long-range dust transport in Europe (Lecce in Italy), and across the Atlantic (La Parguera in Puerto Rico). The time-series show an overestimation in the Control experiment of the optical depth

near the sources, and to a smaller extent in the transport which clearly suggests that the model tends to overestimating dust emissions. The current calibration for model version 1.0 has the shortcoming to accurately capture long-range transport at the expenses of an overestimation over the sources. This overestimation is reduced with data assimilation. By a first eyeball inspection, the AOD simulation variance is reduced by data assimilation and is more in accordance with the AOD observation

variance.

Maps in Figure 11 show results of validation statistics calculated for the full study period at each AERONET station for the three experiments performed. These maps allow us to appreciate the strongest features of the three simulations at individual AERONET stations and how those stations are representative of the regional domains that we have identified. The Control experiment shows

that the strongest BIAS and highest RMSE are in the sub-Sahel region. The BIAS indicates that the model systematically over-predicts AOD in that region. The highest FRGE are in the long transport over the Atlantic or Europe as expected in areas of low AOD values. The correlation between model and observation values is in general lower near source areas than in outflow regions. This could be due to the too coarse model resolution not able to follow as good as the observations the dynamic of

the dust field near source areas. The assimilation of MODIS NRL observations decreases some of the strongest biases in particular in the dust outflow regions in Sahel and over the African Atlantic coast, which is reflected in a reduced FRGE and RMSE, and is associated with improved correlation. The assimilation of the MODIS Deep Blue observations additionally to the NRL MODIS observations



is of further benefit: it reduces the BIAS and RMSE downwind from the strongest dust sources of
520 Sahara. It is also relevant to notice that the additional assimilation of MODIS Deep Blue observations
improves the correlation over the above areas and in the Arabian peninsula.

The chart plots for the validation statistics calculated for all the AERONET stations considered
(hereafter called global statistics) and for stations grouped according to regional domains of interest
are respectively in Figure 12 and Figure 13. Global statistics show that assimilation produces in
general a better representation of dust concentrations in the atmosphere, and that the assimilation of
Deep Blue retrievals has a positive impact over the assimilation of Dark Target retrievals only.

When considering the regional domains, the assimilation of NRL MODIS AOD has a positive
impact on the quality of the analysis everywhere, with the only exception of a slightly increase of
RMSE in the Middle East region. This positive impact is more pronounced in the short Atlantic
transport and in the sub-Sahel region. The additional assimilation of MODIS Deep Blue AOD has
a considerable positive impact in the Sahara, sub-Sahel and Middle East regions, and neutral or
slightly detrimental in the rest of the transport, in particular in the long range Atlantic transport.

It should be noted, however, when interpreting the above statistics that the validation against
AERONET observations introduces errors when comparing a global model grid-box against a point
observation (Schutgens et al., 2016).

### 7.3 Validation of the forecast

We have validated the forecast up to 5 days ahead initialized at 0 UTC from either the control
experiment or an analysis (from DA-NRL-DB). We have calculated for May to August 2007 the
errors for the forecast at 12, 36, 60, 84, 108 hours (hereafter indicated as FC+12, FC+36, FC+60,
FC+84, FC+108) with respect to either AERONET observations or satellite observations. As men-
tioned when describing our evaluation methodology, we use as reference the set of satellite obser-
vations from the Dark Target and Deep Blue algorithm ingested in the assimilation step, i.e quality-
controlled and filtered for dust-dominated conditions. They are used only to validate the forecast
range following the assimilation window. As expected, all the validation statistics worsen with in-
545 creased forecast step in both experiments (see Figure 14 for global statistics). The impact of initializ-
ing the model with a dust analysis is positive in the first day. The analysis produces a better forecast
in terms of BIAS and RMSE (and also SD of the error) up to FC+108, and a better correlation in
the first day. The correlation is slightly lower from FC+36 onwards. The conclusions drawn by val-
idating against AERONET or satellite observations are equivalent. Results calculated for regional
domains (Figure 15) show that the Control experiment tends to overestimate AOD everywhere with
the exception of central and east Asia. This suggests an overestimation in particular of the Sahara
emissions which is consistent with the bias found in the analysis and which is maintained during the
forecast. Initializing the 0 UTC forecast with the DA-NRL-DB dust analysis reduces the overestima-
tion compared to satellite retrievals in the first day of the forecast consistently with the improvement





observed in the analysis in the previous section. However, this produces an underestimation of AOD in the long-range Atlantic transport during all the forecast lead times, which, because of the relatively small AOD values in that area, is reflected in particular in the FGRE. Although there is an overestimation of AOD, there is a better agreement of the temporal evolution in that region. The underestimation of AOD in the Atlantic transport might be due to too strong deposition which affects

in particular the long-range transport, and in the standard run is compensated by an overestimation over the sources. As said earlier, a shortcoming of the current model calibration is to capture well the long-range transport at the expenses of an overestimation over the sources, which data assimilation reduces. To identify the exact cause for it will require, however, further investigation together with a better adjustment of the current model parameters. With the exception of this underestimation

of AOD across the Atlantic, all the error statistics and correlation coefficients are improved in the first day of the forecast in all the regional domains. The error of the analysis-initialized forecast is lower for more than 4 days into the forecast, though, after day 1, the temporal evolution is less in agreement (lower correlation) with satellite observations in some regions (SubSahel and ShortAtl), compared to a standard forecast. It is particularly relevant to notice that the dust forecast over Sahara

is improved for all the statistics and throughout the forecast range.

## 8 Conclusions

We have developed a data assimilation system for the NMMB/BSC-CTM model version 1.0, which considers dust only, while other aerosols are being implemented. We have coupled the NMMB/BSC-CTM with an ensemble-based data assimilation technique known as LETKF. For this purpose we

have created a forecast ensemble based on known uncertainties in the physical parametrizations of the mineral dust emission scheme. We have processed satellite aerosol optical depth retrievals for assimilation with a dust filter. Due to the presence of other aerosols in the selection of dust-dominated conditions, uncertainties might have been introduced in our assimilation process. It should be noted however that the identification of dust-dominated conditions is performed in this study as a proof

of concept to demonstrate the potential of using data assimilation in NMMB/BSC-CTM, and will not be strictly necessary in a future model upgrade including all the major aerosol species. Still, efforts towards aerosol speciation could continue to be pursued when assimilating information about total aerosol optical properties. In this respect, operational centres currently rely merely on model background to distribute assimilation increments among the different aerosol species.

Assimilation experiments showed that aerosol optical depth retrieved with the Dark Target algorithm can help NMMB/BSC-CTM to better characterize atmospheric dust. This is particularly true for the analysis of the dust outflow in the Sahel region and over the African Atlantic coast. The additional assimilation of Deep Blue retrievals has a further positive impact in the analysis downwind from the strongest dust sources of Sahara and in the Arabian peninsula.





An analysis-initialized forecast performs better (lower forecast error and higher correlation) than a standard forecast everywhere in the first day of the forecast. The only exception to this is an underestimation of the forecast of AOD in the long-range Atlantic transport. The error of the analysis-initialized forecast is lower also in the rest of the forecast range (up to 5 days), though, after day 1, in sub-Sahel and short Atlantic transport the temporal evolution of dust is less in agreement with

independent observations, compared to a standard forecast. Particularly relevant is the improved forecast over Sahara throughout the forecast range thanks to the assimilation of Deep Blue retrievals over areas not easily covered by other observational datasets. To the best of our knowledge, this is the first study quantifying the benefit of assimilating MODIS Deep Blue from Collection 6 specifically for mineral dust simulations. This product is currently operationally assimilated by the UK

Met Office who consider only Deep Blue observations over desert, and by the European Centre for Medium-Range Weather Forecasts.

     In our future implementation of the forecast ensemble, we plan to exploit spatial patterns of variation in model parameter uncertainty, for example source-dependent uncertainties, as well as uncertainties in the deposition term. A better representations of uncertainties in dust emission flux

inherently will help the representation of uncertainties in other parts of the dust cycle. A recent study by Rubin et al. (2016) shows that, for their system, a combined meteorology and aerosol source ensembles are necessary to produce sufficient spread in outflow regions. Notwithstanding that their conclusion might be system-dependent, we will be take into account their results in our future studies.

**9   Code availability**

Copies of the code are readily available upon request from the corresponding authors.

*Acknowledgements.*   This work was funded by the SEV-2011-00067 grant of the Severo Ochoa Program awarded by Spanish Government, the CGL-2013- 46736-R grant of the Spanish Ministry of Economy and Competitiveness, and the ACTRIS Research Infrastructure Project of the European Union's Horizon 2020 research

and innovation programme under grant agreement n. 654169. The authors thank all the Principal Investigators and their staff for establishing and maintaining the AERONET sites, NRL-University of North Dakota for the MODIS AOD L3 product, the MODIS and OMI mission scientists and associated NASA personnel for the production of the AOD, AAI and AE data used in this investigation. All the simulations have been run on the MareNostrum supercomputer hosted by BSC.



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



**Table 1.** Characteristics of the simulation runs.

| Experiment name | Ensemble configuration | Dust initial conditions at 0 UTC on day 1 | Spin-up period | Dust initial conditions at 0 UTC after day 1 |
|---|---|---|---|---|
| Control | No | Cold start | 1 month | FC+24 from previous day run |
| ENS-free-run | Yes | Warm start from Control | 11 days | FC+24 of the individual members from previous day run |
| DA-NRL | Yes | Warm start from ENS-Free-run | None | Analysis at 0 UTC of the individual members from previous day DA cycle |
| DA-NRL-DB | Yes | Warm start from ENS-Free-run | None | Analysis at 0 UTC of the individual members from previous day DA cycle |
| AN-initialized | No | Warm start from Control | None | Ensemble mean analysis from DA-NRL-DB |





**Table 2.** Regional domains and respective groups of AERONET stations used for validation purposes

| Regional domain (short name) | AERONET stations |
|---|---|
| Long Atlantic transport (LongAtl) | La_Parguera, White_Sands_HELSTF, Univ_of_Houston |
| Short Atlantic transport (ShortAtl) | Capo_Verde, Dakar, La_Laguna |
| Sub-Sahel (SubSahel) | IER_Cinzana, Banizoumbou, Ilorin, Agoufou |
| Sahara (Sahara) | Tamanrasset_INM |
| Extended Mediterranean (ExtMediter) | Saada, FORTH_CRETE, Lecce_University, Rome_Tor_Vergata |
| | Villefranche, Avignon, Evora, Barcelona, Granada |
| Middle East (MiddleEast) | SEDE_BOKER, Solar Village, Hamim |
| Central Asia (CenAsia) | *None* |
| East Asia (EastAsia) | *None* |

**Table 3.** Statistics of departures of first guess and analysis from assimilated observations, calculated for May to August 2007.

| Experiment (departures) | Observations | BIAS | RMSE | CORR | FRGE | SD |
|---|---|---|---|---|---|---|
| DA-NRL (FG) | NRL | 0.074 | 0.37 | 0.59 | 0.66 | 0.36 |
| DA-NRL (AN) | NRL | 0.118 | 0.27 | 0.75 | 0.54 | 0.24 |
| DA-NRL-DB (FG) | NRL | 0.160 | 0.35 | 0.58 | 0.70 | 0.31 |
| DA-NRL-DB (AN) | NRL | 0.169 | 0.29 | 0.72 | 0.61 | 0.24 |
| DA-NRL-DB (FG) | DB | 0.001 | 0.35 | 0.40 | 0.49 | 0.35 |
| DA-NRL-DB (AN) | DB | 0.075 | 0.23 | 0.64 | 0.35 | 0.22 |

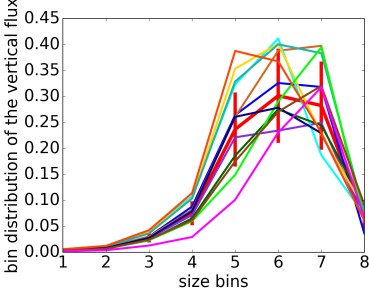

**Figure 1.** Distribution of the mass vertical flux at sources across the eight dust transport bins for the different ensemble members in different colours. The distribution derived from D'Almeida (1987), and used in the standard forecast, is in thick red, with horizontal bars indicating the standard deviation of the noise used to create the perturbations.



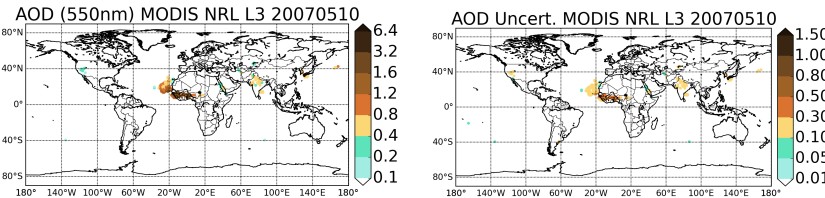

**Figure 2.** Aerosol optical depth (left) and its associated observation error (right) for May 10 2007 for the NRL MODIS Level 3 product after the application of a filter for dust-dominated conditions.

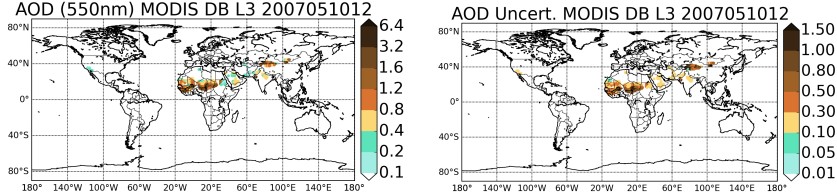

**Figure 3.** Aerosol optical depth (left) and its associated observation error (right) for May 10 2007 for the MODIS Deep Blue Collection 6 Level 3 product after the application of a filter for dust-dominated conditions.

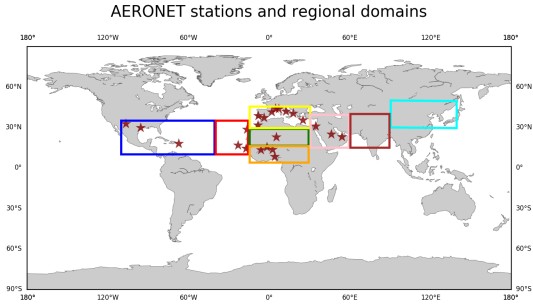

**Figure 4.** Map of AERONET stations and of the different regional domains used for validation purposes.




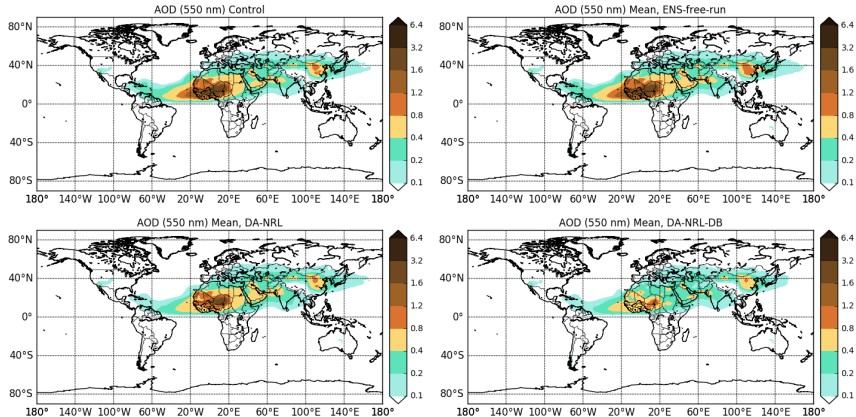

**Figure 5.** Aerosol optical depth averaged for the month of May 2007 for the Control (top left), ENS-free-run (top right), DA-NRL (bottom left) and DA-NRL-DB (bottom right) experiment.

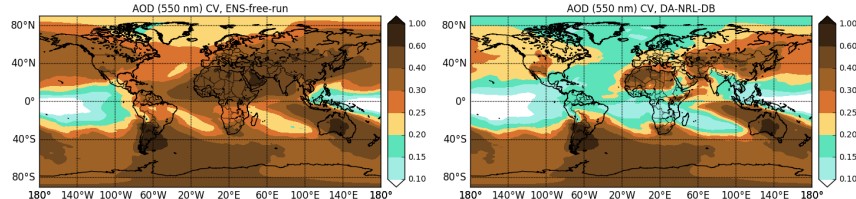

**Figure 6.** Coefficient of variation for the month of May 2007 for the ENS-free-run (left) and DA-NRL-DB (right) experiment, when the ensemble is created perturbing the emitted mass vertical flux for each dust bin and the threshold on the friction velocity generating dust horizontal flux.

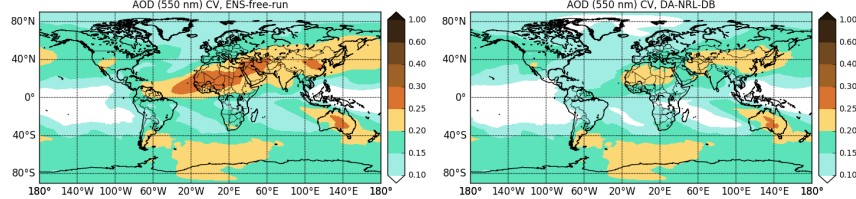

**Figure 7.** Coefficient of variation for the month of May 2007 for the ENS-free-run (left) and DA-NRL-DB (right) experiment, when the ensemble is created perturbing the emitted mass vertical flux for each dust bin.





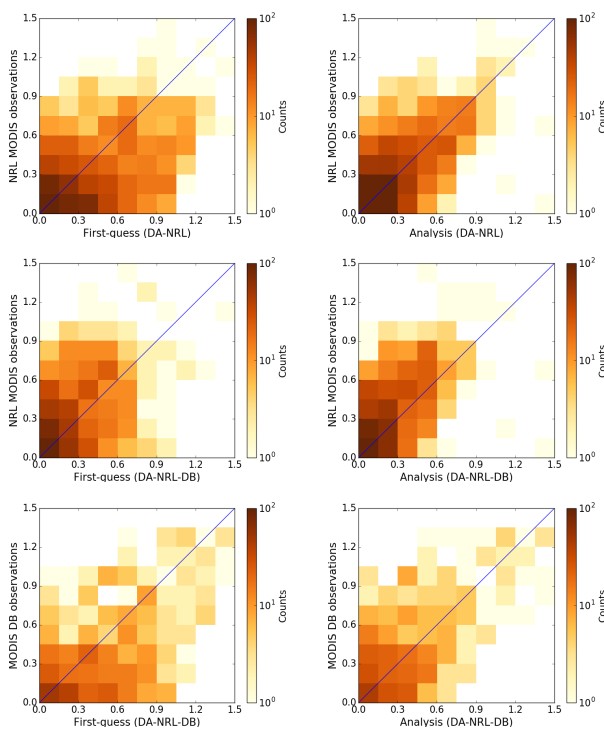

**Figure 8.** Binned scatter plots of the counts of the logarithm of assimilated observations and first-guess (left plot) and analysis (right plot) for the DA-NRL experiment (top row) and DA-NRL-DB experiment (central and bottom rows), calculated for May to August 2007. A logarithmic scale is used for the counts.

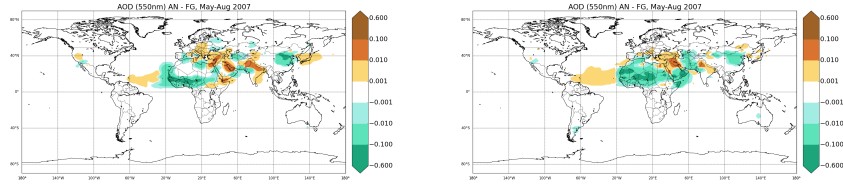

**Figure 9.** Analysis mean increments for May to August 2007 at 12 UTC for the DA-NRL experiment (left) and for the DA-NRL-DB experiment (right).





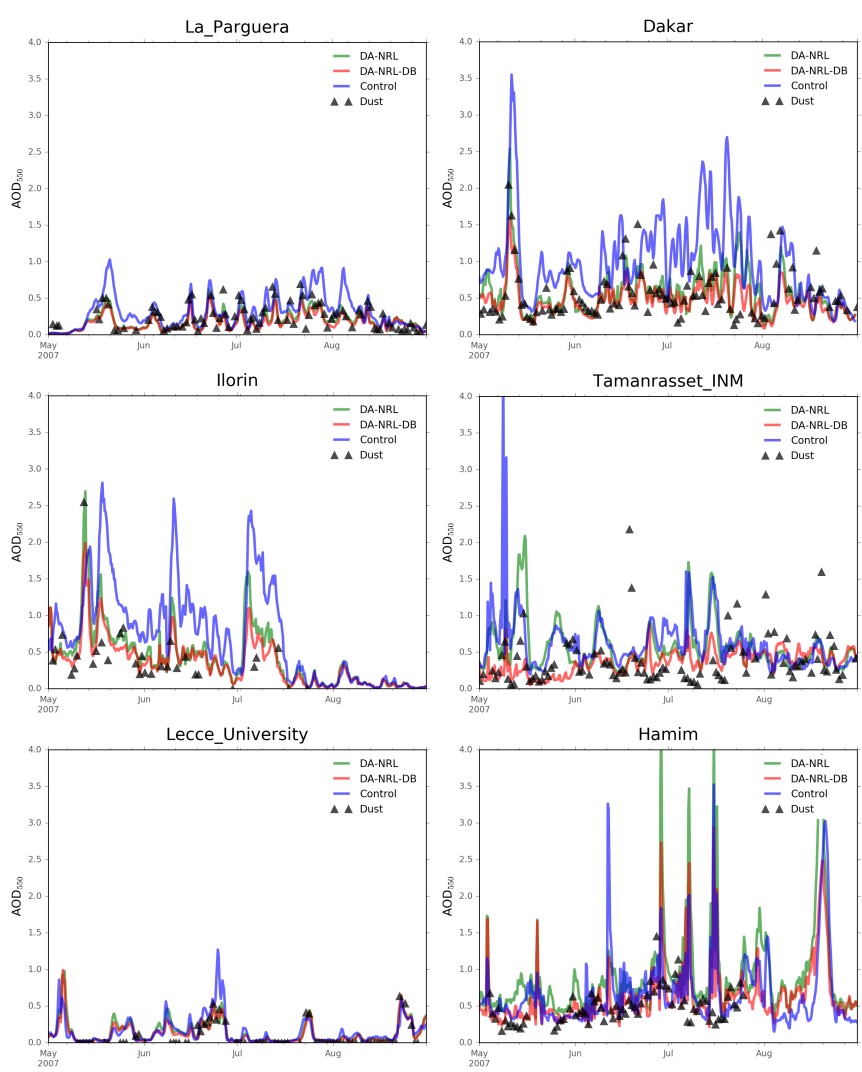

**Figure 10.** Time-series of AOD values for May and August 2007 in La Parguera (top left), Dakar (top right), Ilorin (centre left), Tamanrasset INM (centre right), Lecce University (bottom left), and Hamim (bottom right) for Control (blue), DA-NRL (green), DA-NRL-DB (red) experiment, and for AERONET AOD in dust-dominated conditions (black triangles).





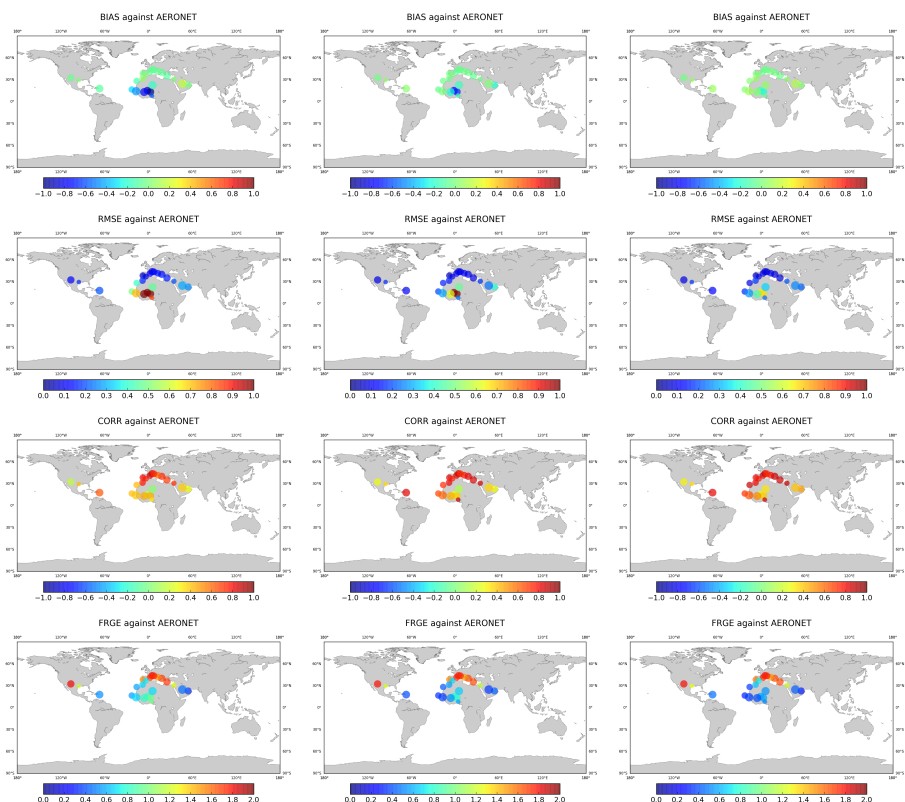

**Figure 11.** Maps of validation statistics: BIAS, RMSE, CORR, FRGE for the Control (left), DA-NRL (centre) and DA-NRL-DB (right) experiment calculated against AERONET AOD for a selection of stations providing observations during the study period (May to August 2007). The size of the circles is proportional to the number of the available samples.




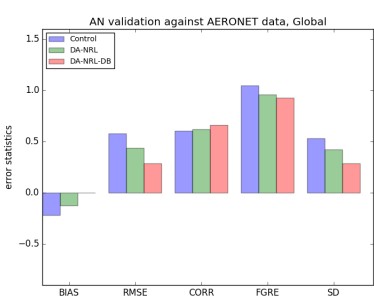

**Figure 12.** BIAS, RMSE, CORR, FRGE and SD for the Control experiment, for the experiment assimilating MODIS NRL observations (DA-NRL) and for the experiment assimilating MODIS NRL and MODIS Deep Blue observations (DA-NRL-DB) calculated against AERONET observations for all the stations in Figure 4.





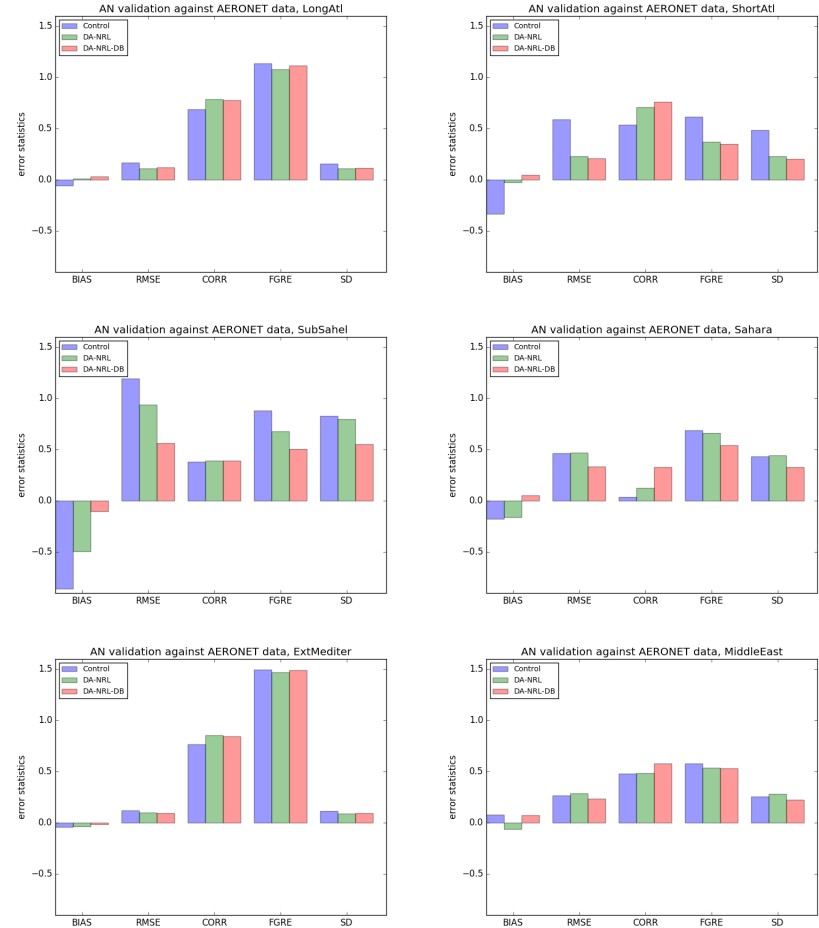

**Figure 13.** BIAS, RMSE, CORR, FRGE and SD for the Control experiment, the DA-NRL experiments and the DA-NRL-DB experiment calculated against AERONET observations for groups of stations within the regional domains in Figure 4.





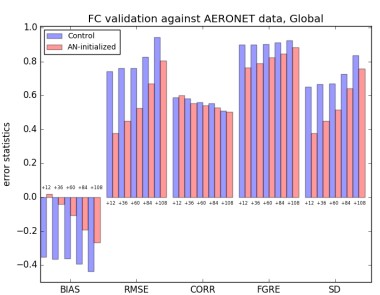
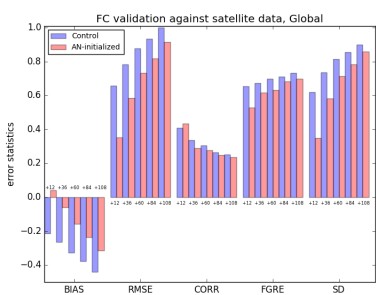

**Figure 14.** BIAS, RMSE, CORR, FRGE and SD for the forecast at 12, 36, 60, 84 and 108 hours of the Control (blue) and AN-initialized (red) experiment, i.e. the experiment initialized with the DA-NRL-DB analysis, calculated against AERONET observations (left) and against global satellite retrievals, both NRL MODIS and MODIS Deep Blues, (right) filtered for dust-dominated conditions. The AERONET stations are the ones in Figure 4.



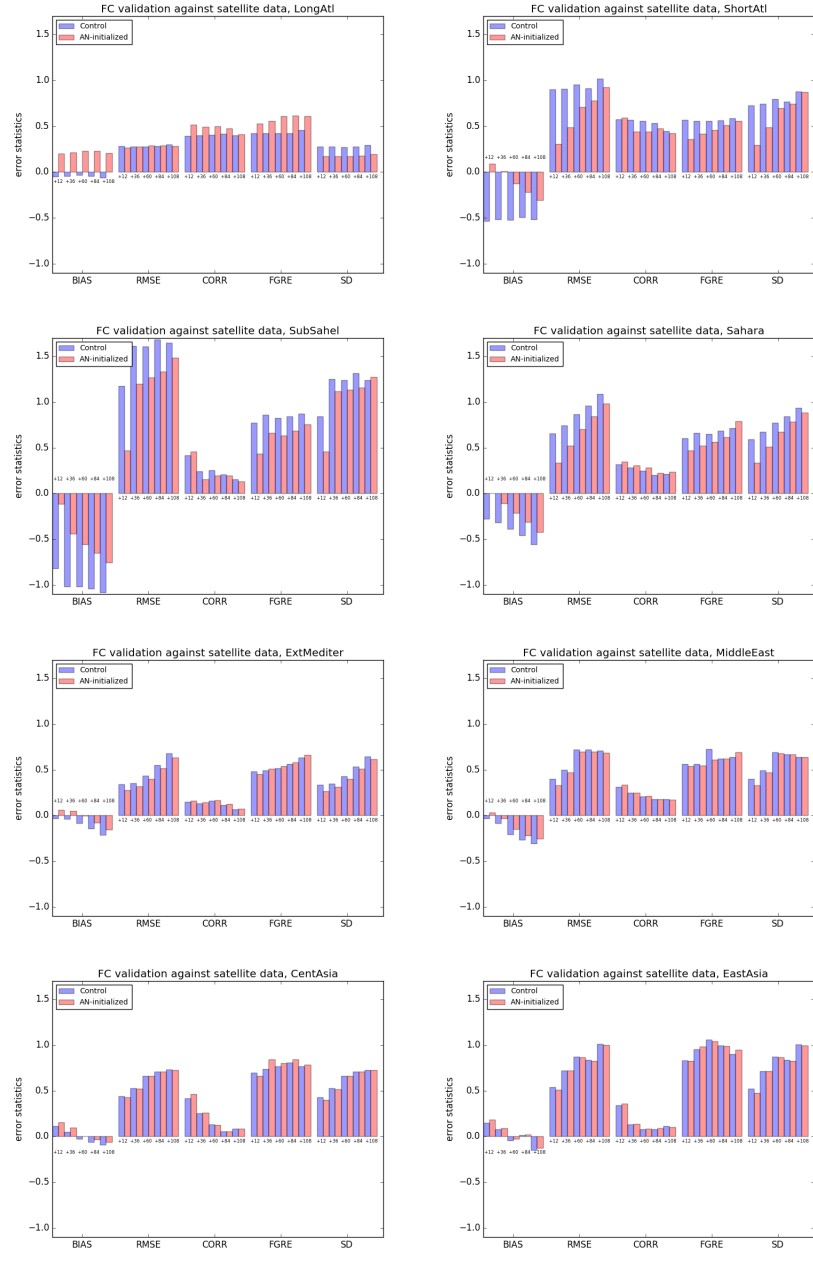

**Figure 15.** BIAS, RMSE, CORR, FRGE and SD for the forecast at 12, 36, 60, 84 and 108 hours of the Control (blue) and AN-initialized (red) experiment, i.e. the experiment initialized with the DA-NRL-DB analysis, calculated in different regional domains against satellite retrievals (both NRL MODIS and MODIS Deep Blue) filtered for dust-dominated conditions.