# Peer review of "Assimilation of MODIS Dark Target and Deep Blue observations in the dust aerosol component of NMMB-MONARCH version 1.0"

_Geoscientific Model Development, 2016_

## Referee Comment (RC1) · Anonymous Referee #1 · 21 Oct 2016

**Review for Assimilation of MODIS Dark Target and Deep Blue Observations in the dust aerosol component of NMMB/BSC-CTM version 1.0**

Overall, I think this is a nice paper that outlines the implementation of an LETKF for constraint of dust aerosol optical depth in the NMMB/BSC-CTM. This work is a first step towards implementing an operational data assimilation system for dust aerosol and, as noted by the authors, multiple species in the future. The paper outlines the dust emission scheme used in the model, which is the basis for generating ensemble members, the LETKF implementation, the observations used in the system for assimilation (MODIS Dark Target and Deep Blue) as well as evaluation of the analysis and forecast. There was clearly a lot of work that went into implementing and testing the new system. I think this paper is worthy of publication, but there a few things that need to be addressed that will add to the clarity of the paper. Overall, I think the figures need to be reworked. They are quite small and the font on the labels is too small to read on most of them, making it harder to evaluate the results. I also have some specific comments and questions below that would help to clarify the results that are presented.

**Specific Comments:**

1. Page 4, Line 98-100, what about the UK Met Office? They assimilate dust AOD in their unified model.
2. Page 4, Lines 107-109, limited spatial correlations have been shown in some studies, depending on what they are sampling, but do you suspect this would be much longer for big dust transport events, especially coming off of the Sahara over the Atlantic ocean? This looks like it would be the case based on MODIS observations.
3. Page 5, I think it would be helpful to include units on the variables in the vertical dust mass flux equation. Also, what value do you use for C? This is constant globally? Is the source mode coefficient how you distribute the mass among the size bins? What threshold do you typically use for the friction velocity (when not perturbed for the ensembles)?
4. Page 5, Line 165, what were the main sources of uncertainty identified in the evaluation efforts?
5. Page 6, to clarify your data assimilation approach, you might want to mention some specifics about the 4-dimensional extension of the LETKF and why you chose to use the extension since you are assimilating observations regularly over 6 hour intervals with the NRL MODIS product. Do you expect to incorporate observations in the future that are asynchronous? Did you test at all the performance of the LETKF versus LETKF with the extension? This would be interesting.
6. Page 7, Line 210-214, what units do you use to define the distance in the localization and what localization factor do you use? It's hard to tell from this how much localization is used.
7. Page 7, Ensemble perturbations in the vertical flux, You are perturbing the distribution of dust emissions among the size bins, but the total mass flux is held fixed? Are the perturbations that you show in Figure 1 the same for all locations or does this vary by grid or region? It might be good to change the solid red line with the error bars in Figure 1 to make it easier to distinguish from the ensemble perturbed lines. Maybe to a dotted or dashed line? Also, it might be useful to show somewhere what sizes the bins correspond to.
8. Page 7, Ensemble perturbations in the threshold friction velocity perturbation. Again, do the random perturbations vary with location or are the same perturbations applied everywhere?

This matters as it will determine your covariances and how an observation spatially impacts your model state.

9.  Page 8, Lines 237-239, if the structure of your source perturbations is temporally and spatially constant, you are essentially specifying your background covariances, much in the way a variational approach operates. As you mention, this is the first stage of development, so I think that's a reasonable first means for generating the ensemble and will probably help you do well near source regions, but you may have problems for transport events.

10. Page 8-9. MODIS Dark Target, I would increase the size of Figure 2, it's too small to see. It would probably also be useful to see some sort of summary of the observations over the experimental time period, perhaps a data count to see where your simulations are being constrained or a mean of your observations. Also, I'm concerned about using over-land AE as a filter for dust. It's been shown that this product is pretty binary (see Levy et al. 2010) and more problematic for coarse mode aerosol than fine. Have you checked to make sure you aren't getting other aerosol in there, like biomass burning aerosol? Perhaps this could be contributing to some of the bias that you are seeing.

11. Page 10, Numerical Setup. The Control is the exact same model as the ensemble free run, the only difference in the ensemble free run is you have perturbed dust emissions (either in the distribution in the bins or threshold friction velocity) and the control is a single run?

12. Page 11, Lines 358-360. I suspect your insensitivity to ensemble size is a result of how you are generating the ensembles themselves (you are sampling from a specified distribution) and also maybe you are heavily localizing (can't tell without units though). This will likely change as you add other perturbations to your system and you may find that you need a much larger ensemble as 12-24 members is quite small.

13. Page 11, Lines 365-367. Did including vertical localization make much of a difference for AOD assimilation? I'm not sure if you tested it without, but I would think that this wouldn't have much impact for a column-integrated observation.

14. Page 12, your use of error as observation minus model is a bit confusing to me. The bias for example would have a negative value when the model is biased high. Typically, you would use your estimator (model) minus the expected value (observation). I would suggest flipping this so that your bias maps in Figure 11 and stat tables/bar graphs don't confuse the reader into thinking the model is biased low when the opposite is true.

15. Page 13, Section 7.1 I think it would be beneficial in Figure 5 to also show the difference between the DA experiments and your ensemble free run (or control). The difference between the DT+DB simulation and the free run is pretty clear, but harder to see with the DT run. Also, I assume this is dust AOD only? If so, you should probably put that in the Figure caption and mention that in the text as well (Page 13, line 444). Are these differences persistent over the entire simulation since you only show one month?

16. Figure 6, Does the DT simulation's coefficient of variation look similar to the DT+DB? If so, you might want to mention that in the text. If they are different, you should probably show both. Also, does the mean AOD change much with the different perturbation schemes (Figure 6 and 7)?

17. In Figure 6, I'm surprised that you have considerable spread in places that I wouldn't expect, like near the poles in the Southern hemisphere. Are the ensemble members being inflated as part of the data assimilation?

18. Page 14, Lines 460-462. This sentence implies the more spread the better since you'll just push towards the observations. However, your goal is to really have sufficient spread that represents the uncertainty in the system. Have you tried to determine whether or not the spread that you are generating is representative of the uncertainty?

19. Figure 8, I would remove the color bars here for each subplot to save space and increase the individual plot size and font size (same for all the figures). I also wonder if you increase the number of bins in your scatterplot, whether the asymmetry that you talk about would be more apparent.

20. Figure 9, The analysis increments that you are showing are in dust AOD? If so, you should add that to the figure caption or labels.

21. Page 15, Section 7.2 For AERONET sites in transport regions, such as La Parguera, it looks like the dust AOD has decreased with data assimilation compared to the control. However, the analysis increments shown in Figure 9 show an increase in AOD. Perhaps the prior state has decreased so much with the near-source corrections that the increase observed over the oceans still produces an AOD at sites impacted by transport that is still less than the control? I'm curious what you found with that.

22. Figure 11, I would put one colorbar at the bottom of each column of figures then maybe add one label at the top of each column (Control, DA-NRL, DA-NRL-DB) and add one label on the y-axis for each row (Bias, RMSE, Corr, FRGE). That way you can increase the size of each map and make the labels larger. Also, it's so small that it is impossible to see any difference in the circle sizes and there is no reference to use to determine what number of samples the circle size corresponds to.

23. Figure 14 and 15 need to be fixed, the labels are way too small to be able to read. It makes it hard to evaluate your forecast results.

24. I wonder if you might want to show in your statistics bar graphs the average dust AOD as well to give some context to how large the errors really are and maybe considering adding error bars (maybe through bootstrapping) to your statistics to test if the differences are statistically significant.

**Technical Corrections:**

1. Page 3, Line 67, change "to different model inter-comparison" to "in different model inter-comparison"

2. Page 3, Line 73, saying the community resorted to data assimilation makes it sound kind of negative. Maybe you could say something like…because of these large uncertainties, the atmospheric composition community has begun to make use of data assimilation for better characterizing and predicting….

3. Page 3, Line 79, you might want to cite the Sessions et al. 2015 paper after the sentence where you mention that assimilation of aerosol observations is now operational at many forecasting centers.

4. Page 10, Line 325.  You should probably cite the AERONET uncertainty

5. As a suggestion on your equations, you may want to go through and make sure the variables are consistent across equations.  For example, in equation 4 the size bins 1 through 8 are indicated with a *b* while in equation 1 they are indicated with a *k*. Later *k* refers to ensemble members. This might confuse the reader.  Also, it would be useful to include units with your variables.

6. Page 13, regions for validation (Lines 430-440).  I think in Figure 4 it would be good to list the regions associated with each box.  You can probably just put this in the figure caption and say which color box goes with which region, to tie the map to Table 2.

7. For Figures 5,6,7, the colorbars are the same on the different subplots within each figure, so I would only show the colorbar once to save space and make the maps larger.  They are too small to see clearly.

8. In the caption for Figure 10, you should mention that this is the analysis AOD and not the prior.

9. Page 17, Line 567-568.  This sentence isn't very clear.  You are referring to the Sahara?  Better temporal evolution, reflected by the increase in correlation with AERONET over time?

---

## Referee Comment (RC2) · Anonymous Referee #2 · 6 Nov 2016

Review of Assimilation of MODIS Dark Target and Deep Blue Observations in the dust aerosol component of NMMB/BSC-CTM version 1.0 by Tomaso et al.

The authors applied the LETKF technique to the chemical transport forecast model (NMMB/BSC-CTM), and performed an assimilation and forecasting experiment for mineral dust with MODIS AOD observations. I found that the manuscript describes the framework of the system and validation results of the experiment very well and is suitable for publication in GMD with minor revision.

General comments:

1:

The abstract is quite prolixity. Abstracts should include only important information.

2:

I cannot read some figures due to poor resolution and small labels. The authors should re-draw the figures.

Specific comments:

Page 6, line 187: The authors use 100 km as the cut-off (localization) length. How do you estimate this values? For example, Rubin et al. (2016) and Yumimoto and Takemura (2011) used more longer length (1000 and about 600 km).

Rubin et al., Atmos. Chem. Phys., 16, 3927-3951, 2016, doi:10.5194/acp-16-3927-2016

Yumimoto and Takemura, Geophys Res. Lett., 38, L21802, doi:10.1029/2011GL049258

Page 7, line 212:

"h" is already used in line 209. Use another character to represent horizontal localization factor.

Section 3.0:

Ensemble-based methods usually use inflation methods. Does this system use any inflation method?

Figure 1:

Can you add ensemble mean of the vertical flux in the figure?

Page 8, line 237-239:

You use AOT (optical column amount) as the observational constraint. How does the system adjust 3D mass concentration fields of dust bins from the 2D observational

constraint?

Page 9, line 274:

Do you consider error in AE? AE over the land may have much large uncertainty than ocean. Can you separate the dust-dominant condition correctly over the land?

Page 9, line 276:

Coverage and observation time of MODIS do not correspond to those of OMI (particularly for AOTs from Terra satellite). How do you derive the AOTs under dust-dominant condition when there is no OMI observation corresponding to? You do not use MODIS measurements from Terra satellite?

Page 11, Line 344:

The authors extend the system to 4D-LETKF. What are the merits of the extension instead of sequential assimilation? You assimilate AOD with 6-hour interval. I read literature suggests that 4-dimentional methods (smoothers) have advantages in assimilating observation with fine temporal resolution comparing with 3-dimentional methods (filters). However, the 6-hour interval is not so short (actually longer for 4D-LETKF). Addition to this, the main purpose of this study is improving of forecasting with assimilation. Why do you choice smoother for this purpose rather than filter? Did you try the 3D-LETKF? Did you find that the 4D-LETKF is superior to the 3D-LETKF in forecast performance?

Page 11, Line 344:

Do you introduce temporal localization? The assimilation window (24 hours) is too long to examine assimilation without the temporal localization.

Page 11, line 365:

The authors use the vertical localization. What are the merit of that for assimilating vertically integrated observations?

Figure 6:

I think this figure shows ensemble spread of dust AOD. Why the spread exhibits much large value all over the Southern hemisphere?

Figure 10:

Could you adjust the vertical axis of panels? For example, AOD values at Lecec_University are too small to plot with vertical axis of 0.0-4.0. Could you add MODIS-measured AOD on the panels? It would be good to see difference (error) in MODIS AOD.

Page 14, line 456:

'Top' should be left. 'bottom' should be right.

Page 14, line 460:

The higher spread does not mean the better spread (background error covariance). If you used the larger perturbation, you'd obtain the higher spread.

Figure 11:

Do you compare model result with AERONET observation in daily average? hourly average? or monthly value?

Figure 9:

There are some regions where the DA-NRL-DB shows opposite increment from the DA-NRL. For example, the DA-NRL-DB obtains negative increment around Somalia Peninsula. However the DA-NRL shows positive one. Does this mean there is biasses between the Dark-target and the Deep Blue AODs?

---

## Author Comment (AC1) · 10 Feb 2017

The authors wish to thank the anonymous reviewer #1 for his/her valuable comments and suggestions.

Please find in the attachment the response to the reviewers' comments and the changes made in the revised manuscript.

Please also note the supplement to this comment:
http://www.geosci-model-dev-discuss.net/gmd-2016-206/gmd-2016-206-AC1-supplement.pdf

---

## Author Comment (AC2) · 10 Feb 2017

**Response to reviewers of: "Assimilation of MODIS Dark Target and Deep Blue observations in the dust aerosol component of NMMB/BSC-CTM version 1.0"**

Enza Di Tomaso, Nick A. J. Schutgens, Oriol Jorba, and Carlos Pérez García-Pando

**General Response**

We wish to thank the reviewers for their interest in our paper, for their constructive comments and useful suggestions that lead to an improved manuscript.

We first note that our model has been recently renamed, after the reviewing process of another GMDD paper on the development of the model gas-phase chemistry component (doi:10.5194/gmd-2016-141). We have substituted in the revised version of the manuscript (also in the title) the name NMMB/BSC-CTM with the new name NMMB-MONARCH, where MONARCH stands for "Multiscale Online Nonhydrostatic AtmospheRe CHemistry model".

In the following, we report our answers to general and specific comments of the reviewers. Reviewers' questions and comments are shown in bold-italic, our answers appear in standard type.

**Specific Response (Anonymous Reviewer #1)**

*Answer to general comment:*

***1. Overall, I think the figures need to be reworked. They are quite small and the font on the labels is too small to read on most of them, making it harder to evaluate the results.***

We have reworked all the figures following the reviewer's suggestion.

*Answer to specific comments:*

***1. Page 4, Line 98-100, what about the UK Met Office? They assimilate dust AOD in their unified model.***

We say at page 18, lines 599-600, of the discussion paper that the UK Met Office assimilates MODIS Deep Blue over desert. Following your comment, we have now added it also in the introduction of the revised manuscript at page 3, lines 93-94.

***2. Page 4, Lines 107-109, limited spatial correlations have been shown in some studies, depending on what they are sampling, but do you suspect this would be much longer for big dust transport events, especially coming off of the Sahara over the Atlantic ocean? This looks like it would be the case based on MODIS observations.***

Correlations in dust AOD are automatically set by the ensemble. However, we limit their use because we use a small patch size. The reviewer has a good point. Spatial correlations in long-range dust transport (in particular for dust fine mode) deserve further investigation. We have added a comment about this in the introduction of the revised

manuscript at page 4, lines 99-102.

**3. Page 5, I think it would be helpful to include units on the variables in the vertical dust mass flux equation. Also, what value do you use for C? This is constant globally? Is the source mode coefficient how you distribute the mass among the size bins? What threshold do you typically use for the friction velocity (when not perturbed for the ensembles)?**

The units for the vertical dust mass flux equation have been added at page 5, lines 145-151 of the revised manuscript. The variables for which we have not specified the units are unit-less. We have added also the value used for the constant *C* (0.768) at line 154, which is an updated value for the global constant estimated in paragraph 4.2.1 of Pérez et al. (2011). *C* is a tuning factor for the total vertical flux. As we say in the discussion paper (page 5, lines 146-148), the total vertical flux mass is distributed among the dust transport bins according to a specific dust distribution at sources derived from D'Almeida (1987) which assumes that the vertical dust flux is size distributed according to three lognormal background source modes. In few words, the mass is distributed at sources among the 8 transport bins according to the coefficients represented with the red thick (red dashed, in the revised version) line in Figure 1. The threshold on the friction velocity is not fixed, but dynamically estimated as function of different soil characteristics, among which soil water content (please see equation 3 to 5 in Pérez et al., 2011). We have now specified also the latter in the revised manuscript at page 5, lines 135-136.

**4. Page 5, Line 165, what were the main sources of uncertainty identified in the evaluation efforts?**

Emissions are identified as one of the sources of uncertainty. Quoting the referred evaluation studies, for example it has been detected an overestimation of the Bodélé emissions and the under estimation of the Mali/Mauritania border emissions. There are however evaluation sites, like Solar Village, where it has not been determined if errors in AOD are due to an inaccurate source prescription or to the inability of the model to reproduce the associated meteorology, or, sites like Izaña, where instead errors in AOD could be due to the deficiency in the model representation of the steep orography. Also, the model tends to overestimate the very low background concentrations far away from sources, which hints towards an overestimation of the smallest dust particles due to either inaccuracies in the size distribution of the emissions, vertical transport and/or removal.

**5. Page 6, to clarify your data assimilation approach, you might want to mention some specifics about the 4-dimensional extension of the LETKF and why you chose to use the extension since you are assimilating observations regularly over 6 hour intervals with the NRL MODIS product. Do you expect to incorporate observations in the future that are asynchronous? Did you test at all the performance of the LETKF versus LETKF with the extension? This would be interesting.**

We have not made the test without the 4D extension. It could be interesting to perform it, however this would require resources (and more I/O time). We are indeed assimilating asynchronous observations with some degree of approximation: the analysis is calculated once a day with observation slices of 6 hours; simulated observations and background departures are calculated at each time slot (every 6 hours) using the background ensemble for that time. As explained at page 11, lines 350-355, of the discussion paper we concatenate observation vectors and background matrices and use the same analysis equation used for synchronous observations. In this, we are following the 4-dimensional

extension of the LETKF described in Section 4 of Hunt et al. (2007). What we have not done, which, quoting Hunt could be of advantage, is to take into account the timing of the observations when deciding which of them to use in a given local analysis.

**6. Page 7, Line 210-214, what units do you use to define the distance in the localization and what localization factor do you use? It's hard to tell from this how much localization is used.**

The distance in the localisation function is calculated in the grid space. We have specified this now in the revised manuscript at page 7, line 209. As we wrote at page 11, line 362, of the discussion paper in the description of the experiment setup, we have set the horizontal localization factor to the value 1, which means that after 2 grid points this function is very close to zero.

**7. Page 7, Ensemble perturbations in the vertical flux, You are perturbing the distribution of dust emissions among the size bins, but the total mass flux is held fixed? Are the perturbations that you show in Figure 1 the same for all locations or does this vary by grid or region? It might be good to change the solid red line with the error bars in Figure 1 to make it easier to distinguish from the ensemble perturbed lines. Maybe to a dotted or dashed line? Also, it might be useful to show somewhere what sizes the bins correspond to.**

Yes, we are perturbing the distribution of dust emission among the size bins, but the total mass flux is not held constant. As we wrote at page 7, line 223-225 of the discussion paper, the model ensemble is created perturbing the vertical flux of dust in each of the eight dust bins. This is equivalent to perturbing the total vertical flux as well as its size distribution at sources.

The source perturbations are constant in time and space as we wrote at page 8, lines 238-239 of the discussion paper.

We have modified the style of the solid red line in Figure 1 (a dashed line in the revised version) to make it more visible, as the reviewer suggested.

We have added the information about bin sizes in the ensemble perturbation section at page 7, lines 220-221 of the revised manuscript and in the caption of Figure 1.

**8. Page 7, Ensemble perturbations in the threshold friction velocity perturbation. Again, do the random perturbations vary with location or are the same perturbations applied everywhere? This matters as it will determine your covariances and how an observation spatially impacts your model state.**

Also these perturbations are spatial and temporally constant. We have now specified at page 7, line 236, of the revised manuscript that what we wrote at page 8, lines 238-239, of the discussion paper refers to both types of source perturbations.

**9. Page 8, Lines 237-239, if the structure of your source perturbations is temporally and spatially constant, you are essentially specifying your background covariances, much in the way a variational approach operates. As you mention, this is the first stage of development, so I think that's a reasonable first means for generating the ensemble and will probably help you do well near source regions, but you may have problems for transport events.**

The spin-up period for the ensemble ensures that perturbations applied at the sources propagate everywhere and dynamically create covariance structures due to the different size distribution, emissions, but also due to observation localization and limited patch size for the local analysis. The background covariances therefore are not constant. However, implementing spatially (and temporally) varying perturbations should be tested in the future in case it can better represents model uncertainty.

**10. Page 8-9. MODIS Dark Target, I would increase the size of Figure 2, it's too small to see. It would probably also be useful to see some sort of summary of the observations over the experimental time period, perhaps a data count to see where your simulations are being constrained or a mean of your observations. Also, I'm concerned about using over-land AE as a filter for dust. It's been shown that this product is pretty binary (see Levy et al. 2010) and more problematic for coarse mode aerosol than fine. Have you checked to make sure you aren't getting other aerosol in there, like biomass burning aerosol? Perhaps this could be contributing to some of the bias that you are seeing.**

We have increased the size and resolution of Figure 2.

We have added a plot of observation counts over the experiment period (Figure 4 of the revised manuscript).

AE over land has indeed considerable uncertain. To overcome this shortcoming in our dust filter we have added a quality control on the assimilated observations based on normalized first-guess departures that rejects observations that are too far from the background (as we say at page 9, lines 304-306 of the discussion paper). We are aware that this is only a temporary solution until we will run the assimilation with a complete aerosol model.

**11. Page 10, Numerical Setup. The Control is the exact same model as the ensemble free run, the only difference in the ensemble free run is you have perturbed dust emissions (either in the distribution in the bins or threshold friction velocity) and the control is a single run?**

Yes, that is correct.

**12. Page 11, Lines 358-360. I suspect your insensitivity to ensemble size is a result of how you are generating the ensembles themselves (you are sampling from a specified distribution) and also maybe you are heavily localizing (can't tell without units though). This will likely change as you add other perturbations to your system and you may find that you need a much larger ensemble as 12-24 members is quite small.**

This hypothesis will have to be tested. There is evidence that the number of ensemble members (if not too small) does not matter too much as long as the model is kept close to a reanalysis.

**13. Page 11, Lines 365-367. Did including vertical localization make much of a difference for AOD assimilation? I'm not sure if you tested it without, but I would think that this wouldn't have much impact for a column-integrated observation.**

Yes, vertical localization does not have much impact without any vertical observational constrain. Since our vertical localization is using the background sensitivity in the vertical,

it is equivalent to distribute the mass increments according to the model background vertical profile.

**14. Page 12, your use of error as observation minus model is a bit confusing to me. The bias for example would have a negative value when the model is biased high. Typically, you would use your estimator (model) minus the expected value (observation). I would suggest flipping this so that your bias maps in Figure 11 and stat tables/bar graphs don't confuse the reader into thinking the model is biased low when the opposite is true.**

We have changed our convention for the model field error which was defined at page 12, line 407, of the discussion paper, now defined with an opposite sign in the revised manuscript at page 13, line 409. We have changed accordingly the sign of the bias in Table 3, Figure 12 to 16 of the revised manuscript.

**15. Page 13, Section 7.1 I think it would be beneficial in Figure 5 to also show the difference between the DA experiments and your ensemble free run (or control). The difference between the DT+DB simulation and the free run is pretty clear, but harder to see with the DT run. Also, I assume this is dust AOD only? If so, you should probably put that in the Figure caption and mention that in the text as well (Page 13, line 444). Are these differences persistent over the entire simulation since you only show one month?**

We have added the difference plots between the DA experiments and the ensemble free run (bottom panels of Figure 6 of the revised manuscript).

Yes, Figure 5 refers to dust AOD only, we have specified this in the text (page 14, line 448) and figure caption (Figure 6) of the revised manuscript.

The difference between the experiments vary during the different months according to differences in the dust emissions and transport over time, but the conclusions stay valid.

**16. Figure 6, Does the DT simulation's coefficient of variation look similar to the DT+DB? If so, you might want to mention that in the text. If they are different, you should probably show both. Also, does the mean AOD change much with the different perturbation schemes (Figure 6 and 7)?**

The DT simulation's coefficient of variation shows higher values in the Northern Hemisphere compared to the DT+DB simulation. These differences are due to less observational constraint over land when DB is not used. We have added the plot (central panel of Figure 7 of the revised manuscript), as suggested, and comments in the manuscript at page 14, line 457 of the revised manuscript.

**17. In Figure 6, I'm surprised that you have considerable spread in places that I wouldn't expect, like near the poles in the Southern hemisphere. Are the ensemble members being inflated as part of the data assimilation?**

No, we have not used inflation. Please note that the plot shows a normalized spread. A considerable normalized spread is expected in the Southern Hemisphere (SH): there the values of dust AOD are quite small, while the ensemble members show differences among them due the perturbation of emissions in the SH sources in South America, Africa and Australia. Prompt by this question, we have double checked how the ensemble spread

evolves from zero on the first day of the spin-up, to values greater than zero only close to the sources in the first days of the spin-up, to finally propagate everywhere in the SH by the end of the spin-up period. We have also added a comment in the text about this at page 14, lines 458-461 since it is a point that raised questions by both reviewers.

**18. Page 14, Lines 460-462. This sentence implies the more spread the better since you'll just push towards the observations. However, your goal is to really have sufficient spread that represents the uncertainty in the system. Have you tried to determine whether or not the spread that you are generating is representative of the uncertainty?**

We have modified the sentence since, as the reviewer correctly pointed out, it implied that the more spread the better, adding the text at page 14, lines 469-477 of the revised manuscript. We have calculated the ratio between prior total spread and RMSE and found that our ensemble configuration is under representing uncertainty. As stated in other part of the manuscript, other perturbations should be tested in the future.

**19. Figure 8, I would remove the color bars here for each subplot to save space and increase the individual plot size and font size (same for all the figures). I also wonder if you increase the number of bins in your scatterplot, whether the asymmetry that you talk about would be more apparent.**

We have used one colour bar for all the sub-plots, increased font size, figure resolution, and also the number of bins in the scatter plots in Figure 9 of the revised manuscript.

**20. Figure 9, The analysis increments that you are showing are in dust AOD? If so, you should add that to the figure caption or labels.**

Yes, the analysis increments are in dust AOD. We have specified it now in the figure caption (Figure 10 of the revised manuscript) and in the text at page 15, line 496.

**21. Page 15, Section 7.2 For AERONET sites in transport regions, such as La Parguera, it looks like the dust AOD has decreased with data assimilation compared to the control. However, the analysis increments shown in Figure 9 show an increase in AOD. Perhaps the prior state has decreased so much with the near-source corrections that the increase observed over the oceans still produces an AOD at sites impacted by transport that is still less than the control? I'm curious what you found with that.**

Yes, it is as the reviewer writes: as a result of the near-source corrections, the overall AOD in La Parguera is less than the control. The analysis increments show local changes while the AOD found in La Parguera is a result of both the local analysis corrections plus the mass transported from Africa which is affected by other local analysis corrections (reduction of mass over Sahara).

**22. Figure 11, I would put one colorbar at the bottom of each column of figures then maybe add one label at the top of each column (Control, DA-NRL, DA-NRL-DB) and add one label on the y-axis for each row (Bias, RMSE, Corr, FRGE). That way you can increase the size of each map and make the labels larger. Also, it's so small that it is impossible to see any difference in the circle sizes and there is no reference to use to determine what number of samples the circle size corresponds to.**

We have used one colour bar for each sub-plot row (as columns represent different statistics with different colour bars), and added one label for each column (experiment) and row (statistics) in Figure 12 of the revised manuscript. We have also added one row for the number of observation samples per stations to have a clearer reference, increased font size and resolution.

***23. Figure 14 and 15 need to be fixed, the labels are way too small to be able to read. It makes it hard to evaluate your forecast results.***

We have increased font size, figure resolution and, to save space, we have removed the bars relative to the standard deviation (SD), in Figure 15 and 16 of the revised manuscript, as it can be derived from the bias and RMSE statistics.

***24. I wonder if you might want to show in your statistics bar graphs the average dust AOD as well to give some context to how large the errors really are and maybe considering adding error bars (maybe through bootstrapping) to your statistics to test if the differences are statistically significant.***

We have added in the plots of the bar graphs in Figure 13 to 16 of the revised manuscript the average value of AOD for the observations  used for validation (indicating the number in one of the upper corner of the plot) and specified this in the figure captions. We have also specified in the text at page 17, lines 554-557 and lines 578-579, whether the results for the correlation are statistically significant.

**Comments on technical corrections:**

***1. Page 3, Line 67, change "to different model inter-comparison" to "in different model inter-comparison"***

We have changed it accordingly, thanks.

***2. Page 3, Line 73, saying the community resorted to data assimilation makes it sound kind of negative. Maybe you could say something like...because of these large uncertainties, the atmospheric composition community has begun to make use of data assimilation for better characterizing and predicting….***

We have changed it accordingly, thanks.

***3. Page 3, Line 79, you might want to cite the Sessions et al. 2015 paper after the sentence where you mention that assimilation of aerosol observations is now operational at many forecasting centers.***

We have added it. Thanks for the suggestion.

**4. Page 10, Line 325. You should probably cite the AERONET uncertainty**

We have added it. Thanks for the suggestion.

***5. As a suggestion on your equations, you may want to go through and make sure the variables are consistent across equations. For example, in equation 4 the size bins 1 through 8 are indicated with a b while in equation 1 they are indicated with a***

***k. Later k refers to ensemble members. This might confuse the reader. Also, it would be useful to include units with your variables.***

We have changed the variable for the size bins in equation 1 of the revised manuscript to have consistency and avoid confusion with the letter used for the ensemble members. We have added the units also for variables used for the AOD operator, unless they are unit-less, at page 8, line 242, of the revised manuscript.

***6. Page 13, regions for validation (Lines 430-440). I think in Figure 4 it would be good to list the regions associated with each box. You can probably just put this in the figure caption and say which color box goes with which region, to tie the map to Table 2.***

We have added it as suggested, thanks.

***7. For Figures 5,6,7, the colorbars are the same on the different subplots within each figure, so I would only show the colorbar once to save space and make the maps larger. They are too small to see clearly.***

We have used one colour bar for the plots sharing the same one in Figure 6, 7, 8 of the revised manuscript and made the maps larger and at a higher resolution.

***8. In the caption for Figure 10, you should mention that this is the analysis AOD and not the prior.***

We have added it as suggested, thanks.

***9. Page 17, Line 567-568. This sentence isn't very clear. You are referring to the Sahara? Better temporal evolution, reflected by the increase in correlation with AERONET over time?***

We have rewritten the sentence at page 18, lines 593-595. We were referring to the SubSahel and ShortAtl regions where the correlation degrades after day 1.

**Specific Response (Anonymous Reviewer #2)**

*Answer to general comments:*

***1. The abstract is quite prolixity. Abstracts should include only important information.***

We have reduced the abstract as suggested, thanks.

***2. I cannot read some figures due to poor resolution and small labels. The authors should re-draw the figures.***

We have reworked all the figures following the reviewer' s suggestion making them bigger, bigger fonts and at a higher resolution.

*Answers to specific comments:*

**1. Page 6, line 187: The authors use 100 km as the cut-off (localization) length. How**

**do you estimate this values? For example, Rubin et al. (2016) and Yumimoto and Takemura (2011) used more longer length (1000 and about 600 km).**

**Rubin et al., Atmos. Chem. Phys., 16, 3927-3951, 2016, doi:10.5194/acp-16-3927-2016**
**Yumimoto and Takemura, Geophys Res. Lett., 38, L21802, doi:10.1029/2011GL049258**

Please see our answer to the specific comment n.6 of reviewer#1. Our cut-off length is hence longer than 100km and in the range of the values used in the studies mentioned by the reviewer. We have added the references suggested to put it in the context of other studies.

*2. Page 7, line 212:*
*"h" is already used in line 209. Use another character to represent horizontal localization factor.*

We have changed the letter for the horizontal localization factor at page 7, lines 208 and 210 to avoid confusion. Thanks for spotting it out.

*3. Section 3.0:*
*Ensemble-based methods usually use inflation methods. Does this system use any inflation method?*

We have not used inflation in the experiments described in the manuscript. We take the reviewer's question as a suggestion for our future tests when with other perturbations, in case the ensemble spread is not representing well enough model uncertainty.

*4. Figure 1:*
*Can you add ensemble mean of the vertical flux in the figure?*

We have added a line (dash-dotted) in Figure 1 for the mean of the ensemble perturbations.

*5. Page 8, line 237-239:*
*You use AOT (optical column amount) as the observational constraint. How does the system adjust 3D mass concentration fields of dust bins from the 2D observational constraint?*

This is explained at page 11, lines 365-367 of the discussion paper.

*6. Page 9, line 274:*
*Do you consider error in AE? AE over the land may have much large uncertainty than ocean. Can you separate the dust-dominant condition correctly over the land?*

We appreciate this concern and in fact use a quality control on the observations. Please see our answer to specific comment n. 10 of reviewer #1.

*7. Page 9, line 276:*
*Coverage and observation time of MODIS do not correspond to those of OMI (particularly for AOTs from Terra satellite). How do you derive the AOTs under dust-dominant condition when there is no OMI observation corresponding to? You do*

***not use MODIS measurements from Terra satellite?***

We use both Terra and Aqua, and, as we wrote at page 8, section 4.1, of the discussion paper, we have used only Level 3, daily, products. When there is no OMI observation, data are not selected for assimilation.

***8. Page 11, Line 344:***
***The authors extend the system to 4D-LETKF. What are the merits of the extension instead of sequential assimilation? You assimilate AOD with 6-hour interval. I read literature suggests that 4-dimentional methods (smoothers) have advantages in assimilating observation with fine temporal resolution comparing with 3-dimentional methods (filters). However, the 6-hour interval is not so short (actually longer for 4D-LETKF). Addition to this, the main purpose of this study is improving of forecasting with assimilation. Why do you choice smoother for this purpose rather than filter? Did you try the 3D-LETKF? Did you find that the 4D-LETKF is superior to the 3D-LETKF in forecast performance?***

Please see our answer to specific comment n. 10 of reviewer #1. It is true however that the literature suggests that it the 4D extension has merits for temporal resolutions finer than the resolution we have used, hence it would be worth testing in any future 3D extension.

***9. Page 11, Line 344:***
***Do you introduce temporal localization? The assimilation window (24 hours) is too long to examine assimilation without the temporal localization.***

No, we have not tested temporal localization for this system. Thank you for the suggestion. Tests by the authors with the LETKF on a different model system have shown no significant difference with a 12 hour window.

***10. Page 11, line 365:***
***The authors use the vertical localization. What are the merit of that for assimilating vertically integrated observations?***

This feature has been implemented to have a system that can handle also the assimilation of profiles, and it is not having impact with integrated observations. Please see our answer to specific comment n. 13 of reviewer #1.

***11. Figure 6:***
***I think this figure shows ensemble spread of dust AOD. Why the spread exhibits much large value all over the Southern hemisphere?***

Please note that the plot shows a normalized spread, which is expected to be considerable on the in the Southern Hemisphere. For more details, please see our answer to specific comment n. 17 of reviewer #1.

**12. Figure 10:**
**Could you adjust the vertical axis of panels?  For example, AOD values at Lecce_University are too small to plot with vertical axis of 0.0-4.0. Could you add MODIS-measured AOD on the panels? It would be good to see difference (error) in MODIS AOD.**

We prefer to use the same vertical axis for the different validation sites to have the

different ranges of AOD values that we are validating in the different regional domains of Figure 4 of the discussion paper, close and far from sources, visually clear.

We have added MODIS AOD from the set of assimilated observations in the time-series of Figure 11 of the revised manuscript. Note, however, that these satellite observations are not an independent reference of validation for the analysis, nor are entirely representative of the observational constraint used to calculate the analysis in the given station location, without taking into account the localisation function and observation uncertainty of all the observation in the local patch around the station location.

**13. Page 14, line 456:**
**'Top' should be left. 'bottom' should be right.**

We have change it accordingly, thanks.

**14. Page 14, line 460:**
**The higher spread does not mean the better spread (background error covariance). If you used the larger perturbation, you'd obtain the higher spread.**

We have modified the sentence following the good point that the reviewer had made. For more details, please see our answer to specific comment n. 18 of reviewer #1.

**15. Figure 11:**
**Do you compare model result with AERONET observation in daily average? hourly average? or monthly value?**

We use the closest AERONET value in a +/- 30 minute interval from the model time step, and we use only one value without doing any averaging. We have specified this in the section Methodology for the evaluation of the simulations at page 13, lines 425-426 of the revised manuscript.

**16. Figure 9:**
**There are some regions where the DA-NRL-DB shows opposite increment from the DA-NRL. For example, the DA-NRL-DB obtains negative increment around Somalia Peninsula. However the DA-NRL shows positive one. Does this mean there is biasses between the Dark-target and the Deep Blue AODs?**

[revised manuscript text omitted]